# Home-field advantage affects the local adaptive interaction between *Andropogon gerardii* ecotypes and root-associated bacterial communities

Anna Kazarina,[1] Soumyadev Sarkar,[1] Shiva Thapa,[2] Leah Heeren,[1] Abgail Kamke,[1] Kaitlyn Ward,[1] Eli Hartung,[1] Qinghong Ran,[1] Matthew Galliart,[3] Ari Jumpponen,[1] Loretta Johnson,[1] Sonny T. M. Lee[1]

**ABSTRACT**   Due to climate change, drought frequencies and severities are predicted to increase across the United States. Plant responses and adaptation to stresses depend on plant genetic and environmental factors. Understanding the effect of those factors on plant performance is required to predict species' responses to environmental change. We used reciprocal gardens planted with distinct regional ecotypes of the perennial grass *Andropogon gerardii* adapted to dry, mesic, and wet environments to characterize their rhizosphere communities using 16S rRNA metabarcode sequencing. Even though the local microbial pool was the main driver of these rhizosphere communities, the significant plant ecotypic effect highlighted active microbial recruitment in the rhizosphere, driven by ecotype or plant genetic background. Our data also suggest that ecotypes planted at their homesites were more successful in recruiting rhizosphere community members that were unique to the location. The link between the plants' homesite and the specific local microbes supported the "home field advantage" hypothesis. The unique homesite microbes may represent microbial specialists that are linked to plant stress responses. Furthermore, our data support ecotypic variation in the recruitment of congeneric but distinct bacterial variants, highlighting the nuanced plant ecotype effects on rhizosphere microbiome recruitment. These results improve our understanding of the complex plant host–soil microbe interactions and should facilitate further studies focused on exploring the functional potential of recruited microbes. Our study has the potential to aid in predicting grassland ecosystem responses to climate change and impact restoration management practices to promote grassland sustainability.

**IMPORTANCE**   In this study, we used reciprocal gardens located across a steep precipitation gradient to characterize rhizosphere communities of distinct dry, mesic, and wet regional ecotypes of the perennial grass *Andropogon gerardii*. We used 16S rRNA amplicon sequencing and focused oligotyping analysis and showed that even though location was the main driver of the microbial communities, ecotypes could potentially recruit distinct bacterial populations. We showed that different *A. gerardii* ecotypes were more successful in overall community recruitment and recruitment of microbes unique to the "home" environment, when growing at their "home site." We found evidence for "home-field advantage" interactions between the host and host–root-associated bacterial communities, and the capability of ecotypes to recruit specialized microbes that were potentially linked to plant stress responses. Our study aids in a better understanding of the factors that affect plant adaptation, improve management strategies, and predict grassland function under the changing climate.

Address correspondence to Sonny T. M. Lee, leet1@ksu.edu.

The authors declare no conflict of interest.

See the funding table on p. 17.

**KEYWORDS** rhizosphere, plant microbiome, ecotypic variation, local adaptation, precipitation gradient

The rhizosphere is a dynamic region characterized by the complex interactions between the plant host and associated microbial communities (1, 2). It has been widely recognized that plants directly and indirectly benefit from associated microbial activities and resultant microbial compounds (3, 4). Complex microbe–microbe interactions around the rhizosphere facilitate nutrient transformations, uptake, and cycling as well as alter soil structure and soil water availability (5, 6). Similarly, plant host–microbe interactions can also greatly affect the overall plant health and productivity (7, 8). Root-associated microorganisms can affect plant resistance to biotic and abiotic stress, help with nutrient uptake and even alter plant morphology and phenology (9, 10). Given the crucial roles of the root-associated bacterial communities, understanding what shapes their assembly, function, and mechanisms, as well as adaptive responses between plant host and associated microbes is critical to predicting the response of this system to changing environmental conditions (11, 12).

Numerous studies have demonstrated the need to consider the interactive effects of the environment and plant host genetics in understanding the root-associated bacterial communities. The chemical and physical characteristics of the local soil can directly affect plant function, which in turn can have consequential influence on the root-associated bacterial composition (11, 13, 14). Plant hosts actively modulate associated microbial communities by releasing various signaling molecules (phytohormones) and compounds into the soil (11, 15). Phytohormones are structurally diverse secondary metabolites released by the plant to activate the immune system (16, 17) in response to microbial pathogens (18) and even insect herbivores (19). In addition to direct immune responses, plants produce phytohormones and compounds in response to abiotic stress such as nutrient or water deficiency, or to promote symbiotic interactions with soil microbes (20). Although there are numerous studies on plant host influence on root-associated bacterial community composition, little information is available in the natural ecosystems about the interactive impact of the host–environment on these communities. The root-associated bacterial community is a subset of microorganisms available in the surrounding soil microbial pool (15). Thus, it is no surprise that the same plants perform differently in distinct locations. In previous studies, the "home-field advantage" is described as stability of the performance of an organism at the "home" environment, and decrease in performance away from "home" (21–23). Interestingly, some studies have described an increase in efficacy of the plant root host-associated bacterial communities' interaction due to the "home-field advantage" (21, 22). However, questions remain if plants locally adapted to the prevailing environmental conditions can take advantage of the local soil microbes or if plants preferably favor microbiomes similar to their home environment.

Due to climate change, drought frequencies and severity are predicted to increase across the United States (24–26). In the Great Plains, drought events limit productivity especially in tallgrass prairies (25). Thus, understanding the effect of drought on shaping the root-associated bacterial communities and the mechanisms of mutual adaptation between plants and associated microbes is critical in the prediction of the ecosystem's response to changing climate. To date, most studies (27–29) on the effect of drought on plant–microbe interactions have been under *in vitro* conditions using model organisms, lacking the complexity and dynamics of the natural systems (30, 31).

To address the question if there is a co-adaptation of the plant host and its associated root-associated bacterial communities in a non-model plant system, our study took the opportunity to investigate the root-associated bacterial composition of three locally adapted big bluestem (*Andropogon gerardii* Vitman) ecotypes planted in reciprocal common gardens across the precipitation gradient in the Great Plains. *Andropogon gerardii* is a perennial $C_4$ grass that is widely distributed across the Great Plains of North America (32, 33), and covers up to 80% of the biomass in tallgrass prairie (32,

34–36). Within the Great Plains, *A. gerardii* has been growing for over 10,000 years along prominent sharp rainfall gradients that range from semiarid to heavy rainfall (37, 38). Time and environmental heterogeneity have lent support for local adaptation of *A. gerardii*, giving rise to distinct ecotypes (dry, mesic, and wet) (37, 39–41). Previous investigations have revealed that *A. gerardii* ecotypes vary in functional traits that influence microbially mediated processes (41, 42). Although there are studies investigating intraspecific (43) and interspecific [reviewed in (41)] plant responses to climate (44), the role of the root-associated bacterial communities in plant host adaptation in the natural system remains unclear.

Here, we (i) investigated the relative importance of the environment and ecotypic variation of *A. gerardii* on establishing the plant root-associated bacterial communities; and (ii) compared the abilities of three regional *A. gerardii* ecotypes planted reciprocally, to recruit microbes in local and non-local environments. We hypothesized that *A. gerardii* ecotypes would perform better in rhizosphere microbial recruitment of unique microbes at the site closely matching their "home" environment, highlighting the effect of the plant genetic background or ecotype on rhizosphere community assembly.

## MATERIALS AND METHODS

### Study sites, samples collection, and processing

We sampled three *A. gerardii* reciprocal gardens in the summer of 2019 before the flowering of the grass. The gardens had been established in 2009 and continually maintained at the three sites: Hays, KS (H) at Kansas State University Agriculture Experimental Station (38°85′N, 99°34′W), Manhattan, KS (MHK) at USDA Plant Materials Facility (39°19′N, 96°58′W), and Carbondale, Illinois (C) at Southern Illinois University Agriculture Research Station (37°73′N, 89°17′W), giving us an excellent opportunity to study the interactive effects of local environment and hosts on the root-associated bacterial communities (Table S1). Experimental details of the reciprocal garden experiment have been published in Galliart et al. (39). Briefly, in 2009, seeds were collected from four populations, which jointly defined each of the three regional ecotypes. Morphological traits of the populations vary among ecotypes, but populations do not differ within ecotypes, thus confirming the regional nature of the ecotypes (38, 40, 45). Seeds were collected from mixed grass prairie in Central Kansas (referred to as CKS/Dry: CDB, REL, SAL, WEB), tallgrass prairie in Eastern Kansas (EKS/Mesic: CAR, KON, TAL, TOW), and Southern Illinois savanna (SIL/Wet: 12MI, DES, FUL, WAL) (acronyms of populations are listed in Table S1) across the natural rainfall gradient with 580 mm/year, 871 mm/year, and 1,167 mm/year of mean annual precipitation, respectively. Selected populations originated from intact prairies within an 80-km radius of each reciprocal garden site (37). Seeds representing the three ecotypes and twelve populations were germinated and grown in a greenhouse using potting mix (Metro-Mix 510). Established 3- to 4-month-old seedlings were then planted at each of the three reciprocal garden sites (size: 4 × 8 m), in which 12 plants (4 populations × 3 ecotypes) were planted in a complete randomized block design with 10 blocks (rows) for a total of 120 plants per site. Plants were planted 0.5 m apart along each row, and the soil around the plants was covered with the water-permeable landscape cloth for weed control. After almost 10 years after establishment in 2009, we sampled *A. gerardii* root-associated bacterial communities *in the* reciprocal gardens for this experiment.

To track microbial communities, we collected the top-most soil layer (top 15 cm) due to the highest microbial activity in this layer of soil. We collected a single core (15 cm deep × 1.25 cm diameter) as close as possible to each plant. The side of the plant was picked randomly to ensure sample heterogeneity. Each core represented an independent sample, i.e., cores were not pooled with other cores. Seven plants had not survived the transplanting or through the 10 years in the common gardens. However, the plant mortality differed in the Pearson's chi-squared test neither for ecotype ($x^2$-squared = 4.44, df = 2, *P*-value = 0.109) nor population ($x^2$-squared = 9.43, df = 12, *P*-value = 0.665),

suggesting that mortality was random without predictable patterns across ecotypes or populations. In addition, we obtained 10 additional soil cores (15 cm deep × 1.25 cm diameter) randomly within each site to characterize the soil pH, texture, and moisture content at the site level. All collected samples were sealed in ziplock bags on site, transported on ice, and stored at −20°C until processed.

We extracted total DNA from 0.150 g of roots and rhizosphere soil from all 353 samples using an Omega E.Z.N.A. Soil DNA Kit (Omega Bio-Tek, Inc., Norcross, GA, USA) as per the manufacturer's protocol with a slight modification. Roots were picked manually from the collected soil cores, shaken to remove non-adhering soil, and weighed for the DNA extractions. Any soil that remained attached to the roots was considered rhizosphere soil (46, 47). We mechanically lysed the cells on a Qiagen TissueLyser II (Qiagen, Hilden, Germany) using glass beads for 2 min at 20 rev/s prior to any downstream DNA extraction steps. Even though the mechanical lysis was a step in the DNA extraction process, the grass roots were never fully homogenized in the process; therefore, our results represent mainly rhizosphere microorganisms and may have excluded many endorhizosphere organisms. The extracted DNA was eluted to a 100-µL final volume. The DNA yield and concentration were measured using a Nanodrop and a Qubit dsDNA BR Assay Kit. Extracted DNA was sequenced (2 × 250 cycles) with the 16S rRNA V4 region amplified using the primers 515F and 806R at the Kansas State University Integrated Genomics Facility. Some of the acquired rhizosphere samples did not produce adequate DNA extracts for metabarcoding or failed to produce metabarcode sequencing data. In our downstream statistical analyses, we used analysis of variance (ANOVA) and its permutational analog (PERMANOVA), which are robust even for heteroscedastic data and non-balanced designs (48), to characterize our data sets due to the uneven numbers of remaining replicates within each site, ecotype, and population. Altogether, our data included a total of 284 *A. gerardii* rhizospheres across the three sites (Table S2).

## Soil chemistry and soil properties analyses

We performed soil %C, %N analysis on an aliquot of rhizosphere soil samples. For soil chemistry, the rhizosphere soil samples were homogenized from each soil core (total $n = 360$) through a 4-mm sieve to homogenize the soil and remove rocks and large pieces of roots and followed by handpicking small roots from each soil sample. Due to the abundance of root material in the sampled cores, the number of weights of some soil samples was not enough to accurately measure the soil chemistry. We did not observe the pattern across low weight samples and therefore removed them from following processing resulting in a total of 278 samples used for the soil chemistry. A 15-g subsample of sieved soil was then dried at 55°C for a week. Soil samples were then homogenized to a fine powder in a mixer mill (SPEX Instruments, Metuchen, NJ, USA) and re-dried them at 55°C. Dried and ground soil (55 mg) were used for the %C and %N measurement using dry combustion followed by gas chromatography on a Thermo Scientific FlashSmart 2000 CN Soil Analyzer (Milan, Italy). To correct %C and %N data for outliers, we removed all the values that were ±2 standard deviations from the mean within each site prior to the analysis. As a result, we removed 34 samples from %C (total samples for %C = 244; H = 79; C = 59; MHK = 106) and 30 samples from %N (total samples for %N = 249; H = 83; C = 59; MHK = 106) (Table S4). The C:N ratio was calculated manually using %C and %N values (total samples for C:N = 244). We used the ANOVA in R Studio for the overall statistical analysis (with site, ecotypes, populations nested within populations, and blocks as factors) as well as the effect of ecotype at each site, followed by pairwise comparisons using Kruskal–Wallis test (49).

The collected soil samples reserved for pH, moisture content, and soil texture (total $n = 30$) were sent to Kansas State University Soil Testing Laboratory (www.agronomy.k-state.edu/services/soiltesting/). The fresh and dry soil weights were recorded to estimate the moisture content. The moisture content was calculated using MC = $(w − d)/w * 100$ formula, where $w$ and $d$ are the wet and dry weight of soil, respectively. Ten grams of soil was used for the pH measurement using the saturation paste method using 1:1 soil

and water ratio. Twenty-five grams of soil was used for soil texture, where the ratios of sand/silt/clay were measured, and the soil textures were identified using the soil texture triangle. We used the ANOVA in R studio for the overall statistical analysis followed by pairwise comparisons using Kruskal–Wallis test (v 4.1.1) (49).

## Sequence data processing and analyses

We used QIIME 2 v. 2021.4 (50) to process a total of 8,353,179 raw sequences, resulting in 5,628,302 bacterial sequences after quality control. We used QIIME 2 plugin cutadapt (51) to remove the primer sequences. Any sequences with ambiguous bases, with no primer, with greater than 0.1 error rate mismatch with primer or any mismatches to the sample-specific 12 bp molecular identifiers were discarded. Following initial quality control, we used DADA2 (52) with the same parameters across two different runs and truncated the reads to length where the twenty-fifth percentile of reads had a quality score below 15 (Forward 231 and Reverse 229). The first run included 24 samples and was used as a trial run for the project, and the rest of the project samples were sequenced when the quality of the samples was confirmed. Since the same primers were used for the first 24 samples, the quality control and primer removal using DADA2 were performed separately. Samples then were merged together (total $n = 284$) and analyzed as one data set. We used the pre-trained SILVA database (v. 138) in QIIME 2 for taxonomic assignment of the bacteria. Sequences were blinded to amplicon sequence variants (ASVs) and any unknown or unclassified ASVs were removed from downstream analysis. We rarefied the data set to 10,000 reads per sample (resulting in 2,104,416 high-quality sequences) to minimize biases resulting from differences in sequencing depth among samples before estimating diversity indices and downstream analyses (53). We used QIIME 2 for the sequence processing pipeline, whereas the subsequent statistical analyses for microbial richness, diversity, and composition were performed using R studio (54).

We used ANOVA to test for main and interactive effects in observed richness ($S_{Obs}$), community (Shannon's H'), and phylogenetic (Faith's PD) diversity of the rhizosphere-associated bacterial communities among the sites [Hays (H), Carbondale (C), and Manhattan (MHK)], ecotypes (dry, mesic, wet), and populations (12MI, CAR, CDB, DES, FUL, KON, REL, SAL, TAL, TOW, WAL, WEB) nested within ecotype. Following overall ANOVA, we used pairwise comparisons using Kruskal–Wallis test to identify factors that were driving the significant effects. To identify the specific group driving the significance, we ran Pairwise TukeyHSD with the adjustment for the multiple comparisons (R studio v 4.1.1) (49).

We used the vegan package (55) to estimate the pairwise Bray–Curtis distances to compare the bacterial communities among the different factors. We then used the ggplot2 package (56) to visualize these data using non-metric multidimensional scaling (NMDS) ordinations. We used a non-parametric PERMANOVA to determine whether bacterial communities differed compositionally among sites, ecotypes, and populations as well as their two- and three-way interactions. In this model, "population" was nested within "ecotype." Following the PERMANOVA, we performed pairwise comparisons using the pairwise.adonis function. We also used betadispr function in the vegan package (55) to test whether the community dispersion differed between any significant groups. To determine if any sites, ecotypes, or ecotypes within each of the sites had ASVs that were disproportionately more abundant in one group than in another, we used indicator species analysis with the "multipatt()" function in R package "indispecies" (v. 1.7.12) (57).

## Oligotyping analyses

We used minimum entropy decomposition (MED) (58) algorithm with default parameters to identify sequence variants among the 16S rRNA amplicon sequences. The MED partitions the sequences into discrete sequence groups by minimizing the total entropy in the data set. We then concatenated the sequences assigned to a specific genus using the Silva (v. 138) reference database and used the supervised oligotyping method

available in the oligotyping pipeline version 2.1 (59). In this supervised oligotyping method, we used Shannon entropy, with a threshold value of minimum 0.2 and minimum substantive abundance threshold of 10, to obtain genus-level oligotypes. We selected *Pseudomonas* and *Rhizobium* oligotypes for further analysis because of their overall high relative abundance across the bacterial ASVs and generated oligotypes [Oligos: Pseudomonadales-Oligos ($n = 6$); Rhizobiales-Oligos ($n = 5$)].

## RESULTS AND DISCUSSION

We analyzed a total of 5,628,302 (19,818 ± 4,531 per sample) high-quality sequences assigned to 1,441 ASVs (phyloseq v 1.42.0 (60), R ape (61). Of the recovered reads, an average of 90% was annotated to the genus level using the Naivé Bayesian classifier (62). Any reads assigned to chloroplasts and mitochondria as well as any unknown or unclassified ASVs were removed from downstream analyses. The bacterial taxon assignments along with their counts in each sample are provided in Table S3. Our data were dominated by the phylum Proteobacteria (45% of all sequences), followed by Actinobacteria (15%), Acidobacteria (12%), and Bacteroidetes (6%). A small proportion of sequences remained unclassified (0.29%). ASV relative abundances were dominated by Proteobacteria (28% of all ASVs), Actinobacteria (14%), Firmicutes (12%), and Bacteroidetes (9%). Similar to sequences, the proportion of unassigned ASVs was small (0.3%).

### Differences in precipitation are associated with distinct soil chemistry and texture parameters across sites

Our overall analysis (with site, ecotypes, populations nested within populations, and blocks as factors) highlighted that site was the main driver of the soil chemistry parameters (ANOVA, pH: $F_{2,27} = 226.31$; $P < 0.001$; Moisture content: $F_{2,27} = 124.90$; $P < 0.001$) (Fig. 1; Table 1; Table S4). As expected, due to the precipitation gradient from east to west, we observed a reciprocal decrease in pH from Hays to Carbondale (H: 7.6 ± 0.11; MHK: 6.96 ± 0.21; C: 5.47 ± 0.29; C vs MHK: $P < 0.001$; C vs H: $P < 0.001$; H vs MHK: $P < 0.001$). We also observed differences in the soil texture across sites. Both Carbondale and Hays were identified as Silt Loam (C: Silt = 73.7%; Clay = 15.1%; Sand = 11.2%; H: Silt = 70.1%; Clay = 17.3%; Sand = 12.6%), whereas Manhattan had more sandy soil and was identified as Sandy Loam (MHK: Silt = 8.2%; Clay = 15.1%; Sand = 57.9%). Consistent with our observations on soil texture and pH, we further showed differences in the moisture content among the sites—Carbondale had the highest moisture content, followed by the Hays and Manhattan. Even though we expected Manhattan to have a higher moisture content than Hays, water retention would not be optimal in the Manhattan sandy soil.

Next, we compared the carbon (%C) and nitrogen (%N) content as well as the C:N ratio in the rhizosphere surrounding soil (ANOVA, %C: $F_{2,222} = 290.03$; $P < 0.001$; %N: $F_{2,225} = 495.31$; $P < 0.001$; C:N: $F_{2,221} = 51.36$; $P < 0.001$) (Fig. 1; Table 1; Table S4). We used pairwise comparison tests and showed that Carbondale was significantly higher in %C and %N than Hays (%C–C: 3.57 ± 0.794; H: 2.89 ± 0.74; %N–C: 0.27 ± 0.05; H: 2.48 ± 0.06; C vs H: $P < 0.001$) and Manhattan (%C–MHK: 1.31 ± 0.34; %N–MHK: 0.10 ± 0.02; C vs MHK: $P < 0.001$), while Manhattan was also lower than Hays in %C (H vs MHK: $P < 0.001$) and %N (H vs MHK: $P < 0.001$). C:N ratio also differed across all locations, and was higher in Manhattan than in Carbondale (C:N–MHK: 14.13 ± 2.193; C: 13.01 ± 0.94; MHK vs C: $P < 0.001$) and Hays (C:N–H: 11.90 ± 1.125; C vs H: $P < 0.001$), while Carbondale was also higher than Hays (C vs H: $P < 0.001$). In contrast to the clear differences among the sites, we observed higher %C and C:N in wet compared to dry ecotype (ANOVA: %C: $F_{2,223} = 3.47$; $P = 0.03$; Dry vs Wet: $P = 0.028$; ANOVA: C:N: $F_{2,223} = 5.29$; $P = 0.005$; Dry vs Wet: $P = 0.011$), while %N was not statistically different among ecotypes across all locations (ANOVA: %N: $F_{2,247} = 2.55$; $P = 0.079$) (Fig. 1; Table 1; Table S4).

Due to the extremely strong site effect, we believe that the differences among dry and wet ecotypes were driven by the sites; therefore, in order to focus exclusively on the effect of ecotype, we split our data set and analyzed them at individual sites. The "homesite" soil characteristics play an important role in plant adaptation to the

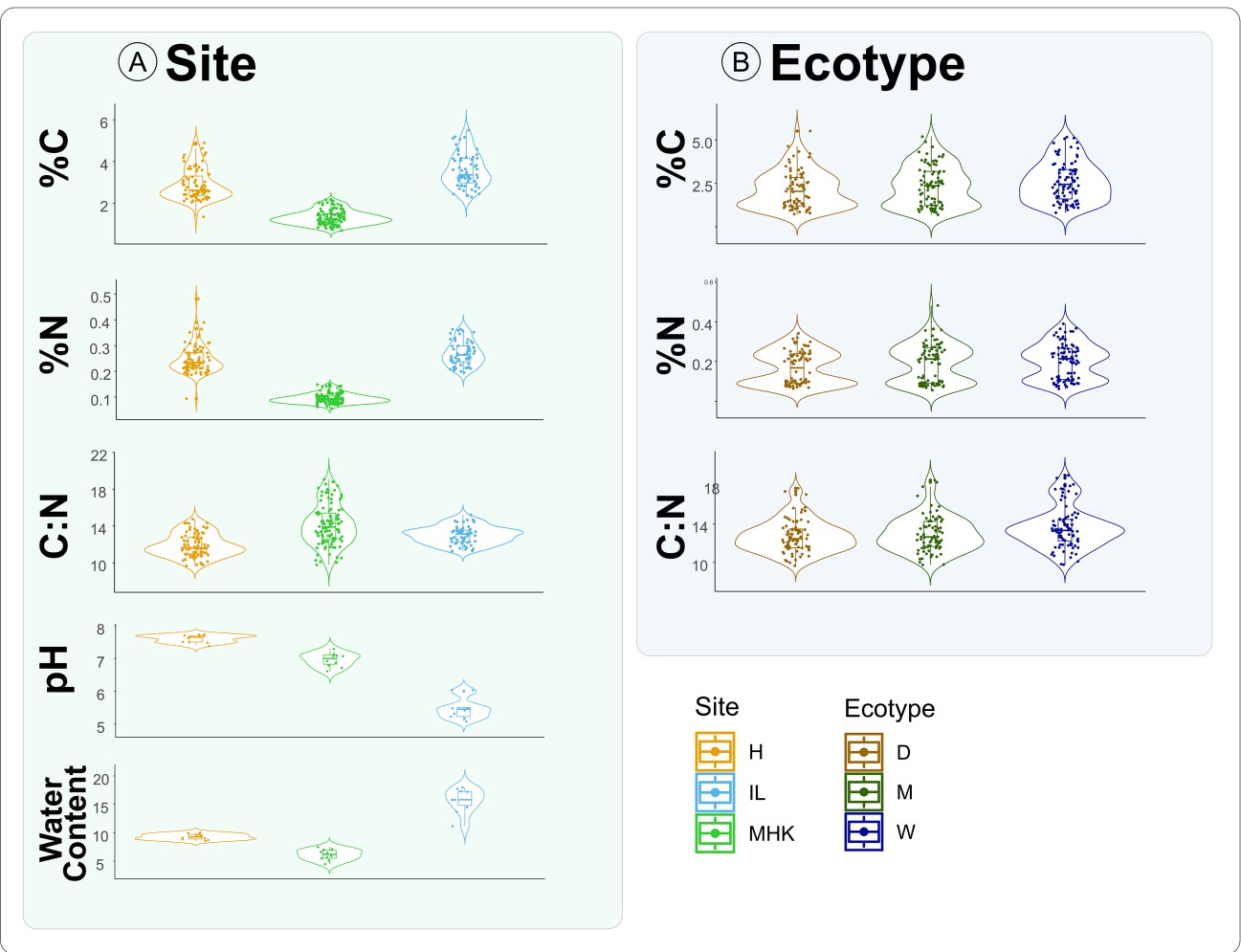

**FIG 1** (A) Soil chemistry and water content among three sites located across the precipitation gradient: Hays, Manhattan, and Carbondale. Soil pH, %C, %N, and the water content. (B) %C and %N concentration across dry, mesic, and wet ecotypes.

environment and ecotypic divergence (37). Therefore, the differences in the ecotypic physiology, such as root length and litter deposit biomass that varied across ecotypes even when growing at the same site, could potentially result in differences of the microbial recruitment (38, 41). In our analysis by site, we observed that the differences in %C were driven by the wet ecotype in Manhattan which was higher when compared to dry and mesic ecotypes (MHK: ANOVA %C: $F_{2,85} = 5.59$; $P = 0.005$; Wet vs Dry: $P = 0.002$; Wet vs Mesic: $P = 0.002$). In addition, we observed that %N and C:N were also significantly different across ecotypes. Similar to %C, differences were driven by the higher values associated with wet ecotype (MHK: ANOVA %N: $F_{2,85} = 7.76$; $P < 0.001$; Wet vs Dry: $P = 0.003$; Wet vs Mesic: $P = 0.003$; C:N: $F_{2,84} = 6.60$; $P = 0.003$; Wet vs Dry: $P = 0.018$) (Fig. 1; Table 1; Table S4). As expected from the combination of higher water content, lower pH, and soil texture factors, Carbondale was associated with the higher %C and %N parameters (63). We surmise that the soil texture was one of the main factors that contributed to the significance of the wet ecotypic soil chemistry difference in Manhattan. The finer soils are associated with the higher stabilization of soil organic matter (SOM) and total %C storage (64); therefore, we did not observe differences between ecotypes in the Silt Loam soil of Carbondale or Hays. In the sandy soil of Manhattan, the differences in the soil carbon deposit would become more distinct between ecotypes and result in higher wet ecotypic carbon deposits due to the wet plant host having a higher overall biomass (64,

**TABLE 1** Statistical analyses (ANOVA and pairwise test) reveal locations (C, Carbondale; H, Hays; MHK, Manhattan) that had a strong influence on the soil chemistry and water content parameters

| Index | ANOVA | | | | | | Pairwise test | | |
|---|---|---|---|---|---|---|---|---|---|
| | | Df | Sum Sq | Mean Sq | F-value | P-value | C | H | |
| pH | Location | 2 | 23.88 | 11.94 | 226.31 | 7.50E-06 | 2.00E-16 | - | H |
| | Residuals | 27 | 1.43 | 0.05 | | | 2.90E-14 | 1.20E-06 | MHK |
| Moisture content | Location | 2 | 474.61 | 237.3 | 124.9 | 2.26E-14 | 6.30E-11 | - | H |
| | Residuals | 27 | 51.3 | 1.9 | | | 5.70E-15 | 2.20E-05 | MHK |
| C% | Location | 2 | 22542.1 | 11271.1 | 290.03 | 2.20E-16 | 7.70E-07 | - | H |
| | Residuals | 222 | 8627.3 | 38.9 | | | 2.00E-16 | 2.00E-16 | MHK |
| N% | Location | 2 | 1.53 | 0.81 | 495.31 | 2.20E-16 | 0.002 | - | H |
| | Residuals | 225 | 0.37 | 0.002 | | | 2.00E-16 | 2.20E-16 | MHK |
| C/N | Location | 2 | 229.07 | 114.53 | 51.37 | 2.20E-16 | 0.00037 | - | H |
| | Residuals | 221 | 491.14 | 2.22 | | | 1.14E-02 | 4.50E-11 | MHK |
| Index | ANOVA | | | | | | Pairwise test | | |
| | | Df | Sum Sq | Mean Sq | F-value | P-value | Dry | Mesic | |
| C% | Ecotype | 2 | 8.922 | 4.4612 | 3.47 | 0.03 | 0.27 | - | Mesic |
| | Residuals | 224 | 13.308 | 1.29 | | | 0.028 | 0.185 | Wet |
| N% | Ecotype | 2 | 0.423 | 0.021 | 2.55 | 7.90E-02 | | | Mesic |
| | Residuals | 227 | 1.88 | 0.008 | | | | | Wet |
| C/N | Location | 2 | 33.52 | 16.76 | 5.29 | 5.68E-03 | 0.408 | - | Mesic |
| | Residuals | 223 | 706.23 | 3.17 | | | 0.011 | 0.55 | Wet |

65). Additionally, although the root C:N strongly differed among the ecotypes across sites (66), our data provided no support for differences in the rhizosphere soil %N among the ecotypes, suggesting that the rhizosphere soils are resistant to the N rhizodeposits, and such changes in the rhizosphere soil may take longer than the decade that the experiment has been in place (67, 68).

## Site had a strong impact on the ecotypic root-associated bacterial communities' richness and diversity

Site was the main driver of the bacterial communities, with a strong effect on bacterial richness and diversity (ANOVA, $S_{Obs}$: $F_{2,245} = 7.13$; $P < 0.001$; Shannon's H' index: $F_{2,245} = 26.56$; $P < 0.001$; Faith's PD index: $F_{2,245} = 4.09$; $P = 0.018$) (Fig. 2A). Pairwise tests showed that community richness ($S_{Obs}$) and Shannon's diversity were lower in Manhattan than in the other two sites (Table 2). On the other hand, the phylogenetic diversity (Faith's PD) in Carbondale was marginally higher than in Hays and Manhattan (Table 2). Phylogenetic diversity analyses suggested that Manhattan was favorable for generalist microbes characterized by the lower diversity (2, 4) due to the intermediate soil chemistry and moisture conditions as compared to Hays and Carbondale. The higher bacterial phylogenetic diversity we observed in this study could be highly attributed to the differences in amount of precipitation and edaphic properties among the sites. Apart from the main effects, we also observed some significant interaction terms in the overall model. Although, the significant interaction terms were contributed to the location and ecotype effects on community richness and phylogenetic diversity (ANOVA, $S_{Obs}$: $F_{10,245} = 2.83$; $P = 0.025$; Faith's PD index: $F_{10,245} = 2.72$; $P = 0.0304$), our pairwise comparison tests did not identify the significant combinations (Pairwise TukeyHSD, $S_{Obs}$: $P_{adj} > 0.283$; Faith's PD index: $P_{adj} > 0.280$).

Similar to the alpha-diversity, the bacterial communities clustered by site indicate that sites had strong effects on bacterial composition (PERMANOVA, $R^2 = 0.42$; $F_{2,245} = 102.53$; $P = 0.001$; Fig. 2B). Our pairwise comparisons further corroborated that all sites had significantly different bacterial compositions (Pairwise Adonis, C vs H: $R^2 = 0.435$; $P = 0.001$; $P_{adj} = 0.003$; C vs MHK: $R^2 = 0.313$; $P = 0.001$; $P_{adj} = 0.003$; H vs MHK: $R^2 = 0.274$; $P = 0.001$; $P_{adj} = 0.003$). We also conducted a dispersion analysis, and observed that bacterial communities in Manhattan ($F_{2,279} = 3.899$; $P = 0.015$) were more heterogeneous

**TABLE 2** Statistical analyses (ANOVA and pairwise test) reveal locations (C Carbondale; H, Hays; MHK, Manhattan) that had a strong influence on the rhizobiome's diversity and richness

| Index | ANOVA | | | Pairwise test | | |
|---|---|---|---|---|---|---|
| | chi-squared | $F_{(2,245)}$ | $P$-value | C | H | |
| SObs | 10.42 | 7.13 | 0.005 | H = 2.70; P = 0.1002 | - | **H** |
| | | | | H = 9.35; P = 6.70e-o3 | H = 4.08; P = 0.0653 | **MHK** |
| Shannon's H | 25.171 | 26.56 | 3.42e-o6 | H = 0.14; P = 0.7 | - | **H** |
| | | | | H = 16.63; P = 6.90e-o5 | H = 21.00; P = 1.40e-o5 | **MHK** |
| Faith's PD | 6.7829 | 4.09 | 0.03366 | H = 4.7; P = 0.044 | - | **H** |
| | | | | H = 5.47; P = 0.044 | H = 0.30; P = 0.585 | **MHK** |
| Index | ANOVA | | | Pairwise test | | |
| | chi-squared | $F_{(2,245)}$ | $P$-value | Dry | Mesic | |
| SObs | 3.46 | 3.46 | 0.033 | H = 0.30; P = 0.585 | - | **Mesic** |
| | | | | H = 3.80; P = 0.078 | H = 6.86; P = 0.026 | **Wet** |
| Shannon's H | 0.91 | 0.91 | 0.404 | H = 1.15; P = 0.285 | - | **Mesic** |
| | | | | H = 1.12; P = 0.285 | H = 5.40; P = 0.060 | **Wet** |
| Faith's PD | 2.51 | 2.51 | 0.108 | H = 0.08; P = 0.779 | - | **Mesic** |
| | | | | H = 2.79; P = 0.095 | H = 4.10; P = 0.130 | **Wet** |

than in Carbondale ($P = 0.020$). Similar to richness, the variation in bacterial community compositions among the sites was likely a result of precipitation-associated biotic and abiotic differences (69) as well as interspecific microbial competition (2, 4). For example, the mean precipitation in Carbondale is 1,167 mm/year, double that in Hays (580 mm/year). This precipitation gradient as well as differences in the soil chemistry and texture among the three sites (Fig. 1; Table S4) would result in differences in pH which is a key factor influencing bacterial diversity (70, 71) (Table 2). Here, we showed that the site had a strong effect on bacterial diversity and composition. We next address if there are any bacterial populations that differed in relative abundance among the sites.

Our indicator taxon (alpha = 0.005) analyses identified 194 taxa that differed among the sites. The majority of the taxa were associated with the wettest and driest sites [Carbondale ($n = 181$); Hays ($n = 151$)], whereas fewer were associated with the mesic intermediate site [Manhattan ($n = 38$)] (Table S5), suggesting that Carbondale and Hays harbored higher number of habitat specialist microbes due to the prevailing environmental conditions (72, 73). Of our three common gardens, Manhattan had the most intermediate conditions for growth of *A. gerardii*, where plants were less likely to experience abiotic stress including water availability stress in Hays and waterlogged conditions in Carbondale. We further observed that the majority of soil bacterial indicator ASVs were assigned to the following five phyla: Acidobacteria (total $n = 31$; C = 21, H = 5, MHK = 5); Actinobacteria (total $n = 66$; C = 15, H = 47, MHK = 4); Planctomycetota (total $n = 20$; C = 13, H = 7, MHK = 0); Proteobacteria (total $n = 109$; C = 53, H = 41, MHK = 15); and Verrucomicrobia (total $n = 23$; C = 14, H = 9, MHK = 0) (Table S5). It is unsurprising that these phyla dominated our indicator taxon analysis since they are ubiquitous among rhizosphere-associated bacterial communities (42, 74–76). However, these indicators were not evenly distributed across sites, suggesting that this pattern was due to the soil moisture affecting the bacterial distribution. Soil moisture can directly affect the soil microbial composition and functionality by differentiating drought tolerance among taxonomic and functional groups of microorganisms (77, 78). For example, limited soil moisture restricts the solute mobility and therefore decreases substrate supply to the soil microbes. Therefore, highly moisture-sensitive Gram-negative bacteria populations (Acidobacteria, Planctomycetes, Verrucomicrobia) were affected by the limited water, resulting in lower number indicators in Hays and disproportionately higher in Carbondale (79–81). In line with this argument, we also observed that Gram-positive bacteria (Actinobacteria) were disproportionately more abundant in Hays than in Carbondale and Manhattan (82).

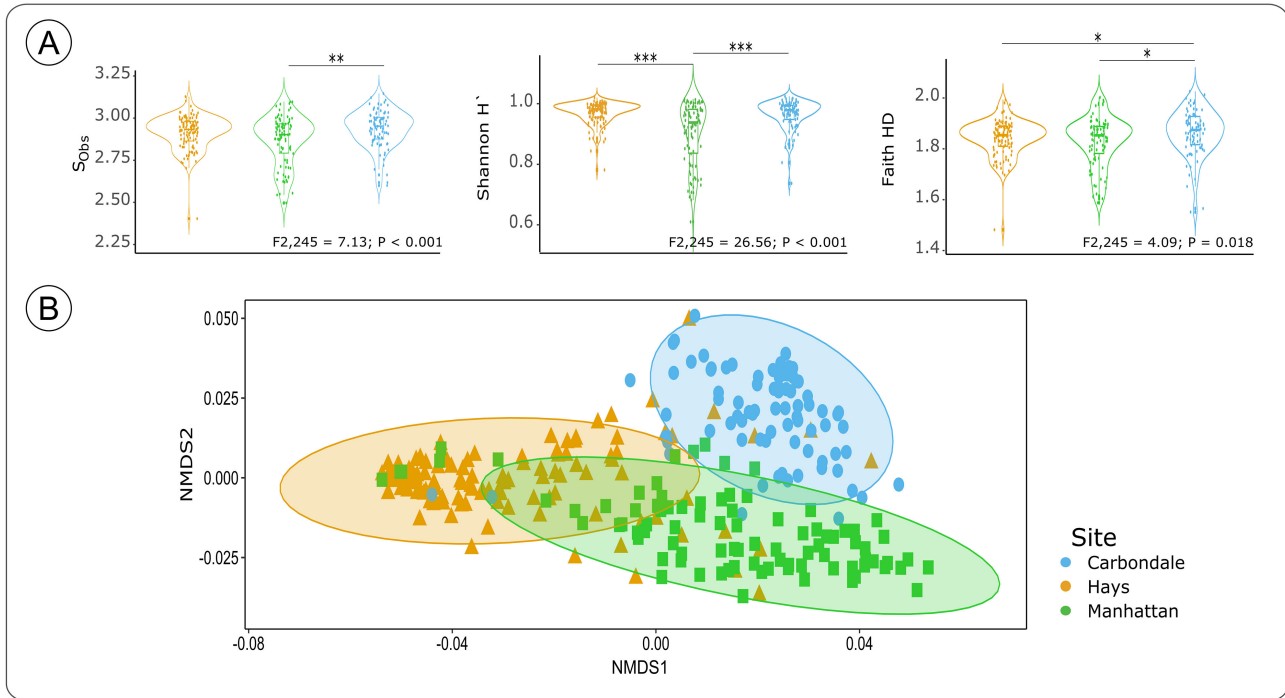

**FIG 2** (A) Bacterial α-diversity indices among three sites located across precipitation gradient: Hays, Manhattan, and Carbondale. Species observed richness ($S_{Obs}$), Shannon's H index, and Faith-PD index. (B) Location impact on *Andropogon gerardii* rhizosphere bacterial composition. NMDS plot of *A. gerardii* rhizosphere communities associated with three site locations across the precipitation gradient: Hays, Manhattan, and Carbondale. NMDS ordinations were obtained from Bray–Curtis similarity matrix (P-values: * −0.05, ** −0.01, *** −<0.01).

## Ecotypes shaped the rhizosphere inhabiting bacterial richness and diversity

While there was a strong site effect on all alpha diversity metrics, a marginal ecotype effect was observed only in ($S_{Obs}$) richness (ANOVA, $S_{Obs}$: $F_{2,245} = 3.46$; $P = 0.033$). The pairwise comparisons among ecotypes indicated that the mesic ecotype host–root-associated bacterial communities had a higher observed richness ($S_{Obs}$) than the wet ecotype (Mesic vs Wet: $H = 6.86$, $P_{adj} = 0.026$). We observed no differences in bacterial diversity and phylogenetic diversity among host ecotypes (ANOVA, Shannon's H' index: $F_{2,245} = 0.91$; $P = 0.404$; Faith's PD index: $F_{2,245} = 2.51$; $P = 0.108$) (Fig. 3A). It was not surprising that the ecotype effect was lower than the site effect. Plant ecotypic effects can often be suppressed by the local biotic and abiotic environmental factors. For example, the environmental gradient defines the local microbial source pool available to the plant and therefore directly affects the source rhizosphere communities (18, 83–85). Taking into consideration the overwhelming effect of the local environment (site), we aimed to test if our ecotypes differed in bacterial richness ($S_{Obs}$) locally. To do this, we split our data set by sites, and performed separate ANOVAs for each site with ecotype, population, and their interaction as the explanatory factors. We used this model separately for Carbondale, Hays, and Manhattan, to focus on the effect of ecotypes at home and away from home sites. Following that, we observed that only in Hays ($F_{2,87} = 15.06$; $P < 0.001$), but not in Manhattan ($F_{2,87} = 0.10$; $P = 0.905$) or Carbondale ($F_{2,79} = 0.41$; $P = 0.668$), did the ecotypes differ. We further used pairwise tests and identified that the wet ecotype in Hays had significantly lower richness ($S_{Obs}$) compared to dry (Wet vs Dry: $P_{adj} < 0.001$) and mesic (Wet vs Mesic: $P_{adj} < 0.001$) ecotypes. These results highlight the potential differences in ecotype recruitment and structuring of the rhizosphere communities even when planted in a common environment. The overall effect of ecotype on bacterial community composition was also weaker than the site effect in the overall model

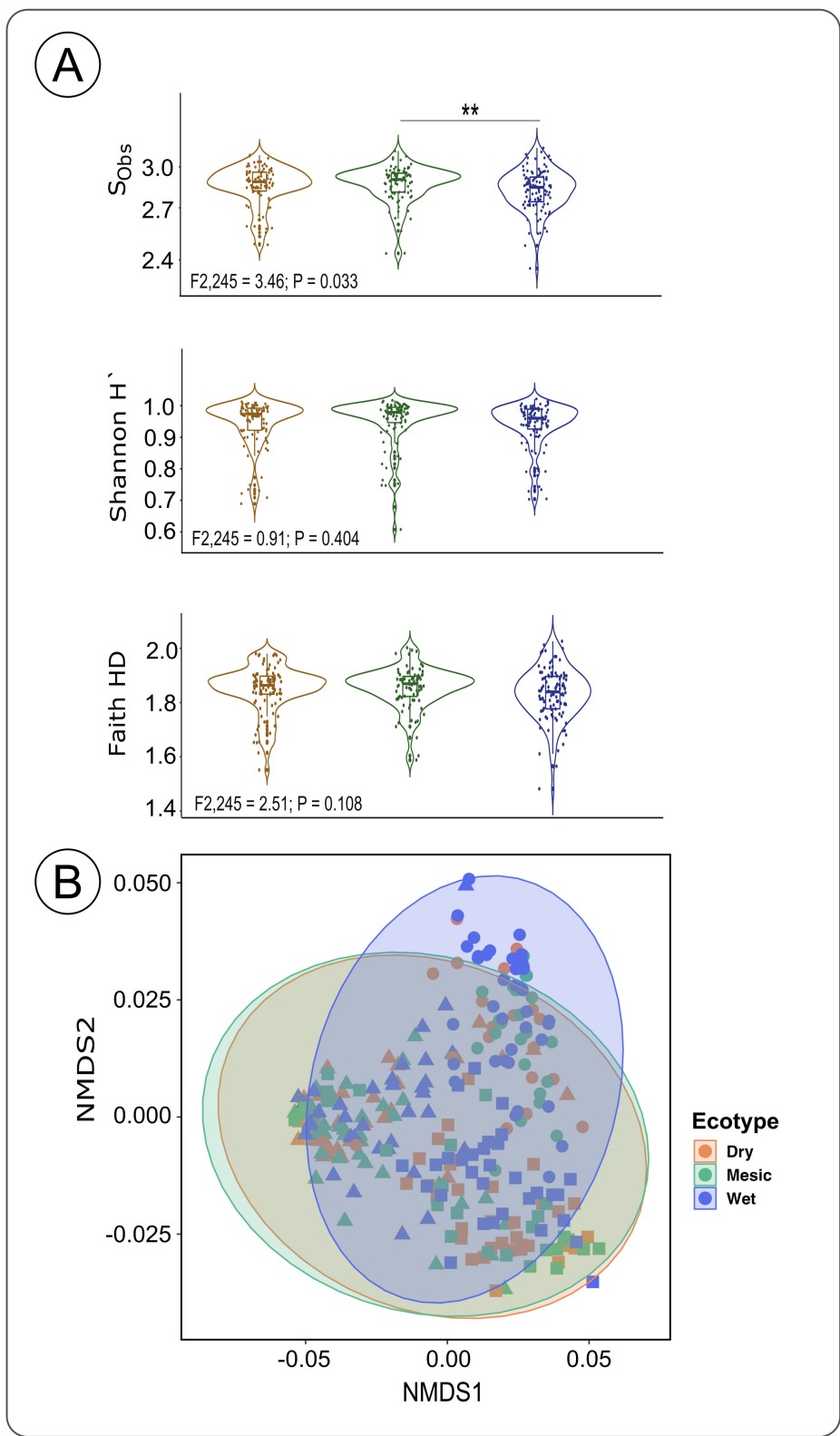

**FIG 3** (A) Bacterial α-diversity indices among the dry, mesic, and wet ecotypes across all locations. Species observed richness ($S_{Obs}$), Shannon's H index, and Faith-PD index. (B) Ecotype impact on *Andropogon gerardii* rhizosphere bacterial composition. NMDS plot of *A. gerardii* rhizosphere communities associated with three ecotypes across three site locations. NMDS ordinations were obtained from Bray–Curtis similarity matrix (*P*-values: * −0.05, ** −0.01, *** −<0.01).

(PERMANOVA, $R^2 = 0.01$; $F_{2,245} = 2.91$; $P = 0.012$). We used pairwise comparisons and showed that the difference among the ecotypic bacterial composition was driven by the wet (Wet vs Dry: $R^2 = 0.028$; $P_{adj} = 0.015$) and mesic ecotype (Wet vs Mesic: $R^2 = 0.038$; $P_{adj} = 0.015$; Fig. 3B).

When we considered the impact of ecotypes on the unique microbial populations at each site, interestingly, we noticed that dry and wet ecotypes harbored more unique taxa when they were planted at their "home field," except for mesic ecotype planted in Manhattan (C: Dry = 79, Mesic = 39, Wet = 127), (H: Dry = 121, Mesic = 71, Wet = 49) (MHK: Dry = 77, Mesic = 67, Wet = 72) (Fig. 4C). Putting together our results on microbial community and indicator taxon analyses, we surmised that interaction between plant ecotypes and sites resulted in the wet and dry ecotypes harboring a higher number of habitat specialist microbes in Carbondale and Hays (72, 73). While the dry ecotype was well-suited to its native environment in Hays, the wet ecotype bacterial communities differed, suggesting that the wet ecotypic communities were not well-adapted to an environment (Hays) that is extremely different from its native location (Carbondale). Therefore, with a mismatch of plant host and root-associated bacterial communities in Hays, the wet ecotype would not be able to get a "home-field advantage," resulting in a higher number of generalists—abundant soil microbes that are good at colonizing plants (2, 4, 73, 86, 87). Another limiting factor for the wet ecotype to recruit and retain native specialist microbes in Hays is driven by lower drought stress tolerance of the plant host (37). Due to the physiological local adaptation to the wetter environment, wet ecotype might be poorly adapted to the drier environment of Hays—resulting in the lower photosynthetic rate (37). We surmise that lesser volume of photosynthetic products exuded in the soil by the wet ecotype in Hays would limit the microbial taxa the wet ecotype could support in the drier environment (88). In addition, the limited photosynthates exuded by the wet ecotype would then be metabolized by the more abundant (73) and faster colonizing generalists (89, 90). On the other hand, the dry ecotype in Hays produced lesser biomass but maintained higher photosynthetic rates, which potentially could result in continuous and steady supply of photosynthate in limited quantities favoring slow growing microbial specialists (73). Due to the general lower relative abundance of specialists in the samples, statistical analysis on ecotypic Shannon's diversity and Faith PD as well as indicator taxon might not be sensitive enough to show the impact of ecotypes on its associated root-associated bacterial communities (91–93).

In order to test this idea, we calculated how many microbial taxa were uniquely observed in each site, to provide insights into the environmental and functional selection of potential specialists' microbial populations. We showed that Carbondale (133) harbored the highest number of unique microbial taxa, followed by Hays (120) and Manhattan (79) (Fig. 4A; Table S6). We then looked at the unique taxa associated with ecotypes across all sites. Wet ecotype had the highest number of unique microbes (107), followed by Dry (92) and Mesic (66) (Fig. 4B; Table S6). To confirm for the independence of the unique microbe group assignments we additionally run the Pearson's $\chi^2$ test for both site ($P > 0.05$) and ecotypes ($P > 0.05$). Our results highlighted that ecotypes gained a "home-field advantage" when planted in their native environment, and were able to match the plant host ecotypes to recruit and retain a higher number of unique microbial populations. On the other hand, the mesic ecotype was consistently associated with a lower number of unique microbial taxa. We hypothesize that the intermediate environment for *A. gerardii* in Manhattan resulted in harboring general microbial taxa that could be equally associated with all the ecotypes, however lacking the unique microbial drivers. Therefore, mesic ecotype was less adapted to recruit microbial specialists, and was associated with a lower number of unique taxa across all the sites.

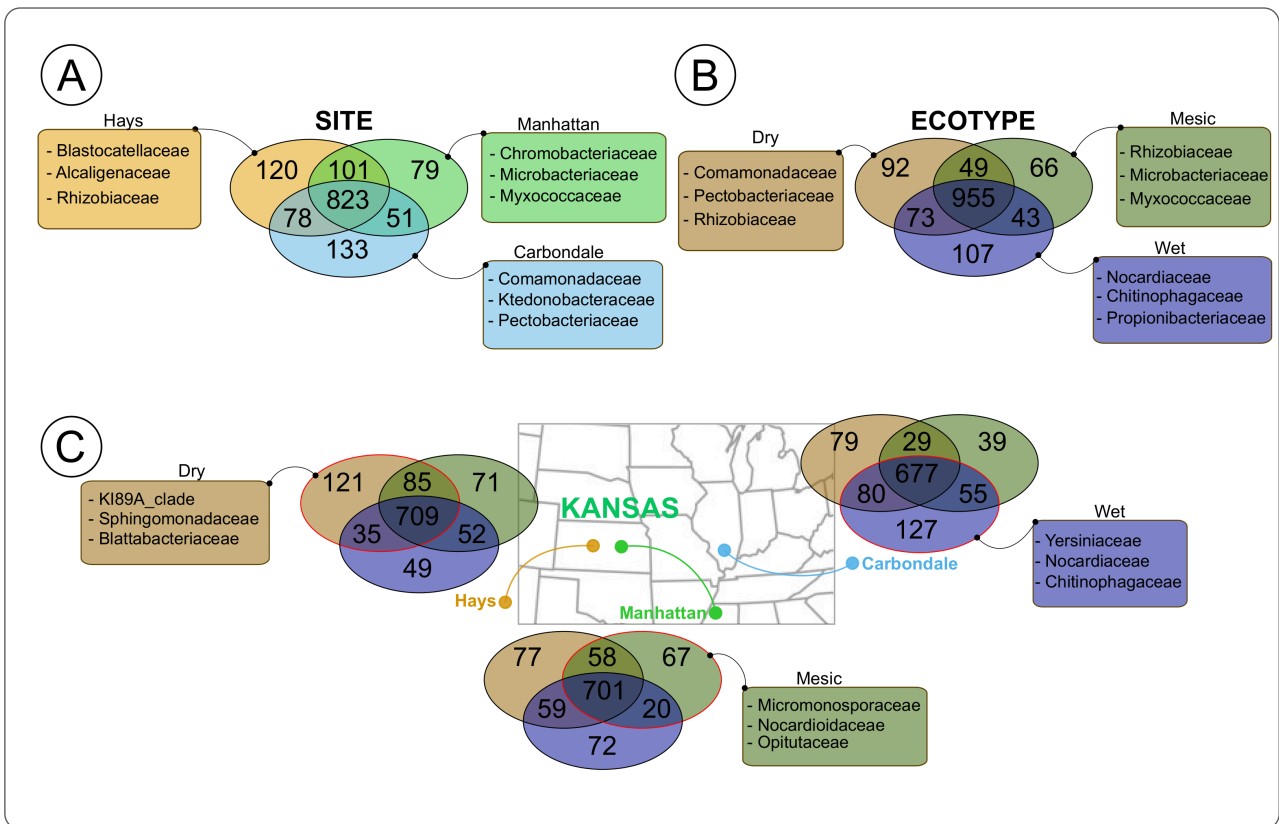

**FIG 4** Venn diagrams representing the overlapping and unique ASVs (numbers in the circle) among (A) all locations, (B) all ecotypes, and (C) ecotypes in Carbondale, ecotypes in Hays, and ecotypes in Manhattan. The circle with a red edge represents the home ecotype for that location. Inserts represent the top three families. Partial information on unique ASVs is shown here, full information is in Table S6. Higher numbers of unique ASVs suggest the "home-field advantage" of the *Andropogon gerardii* ecotypes.

## Matching of the plant host ecotype and root-associated bacterial communities was evident at the ecotype homesite

We used indicator taxon analysis to give insights into the interactive relationship between the ecotypic plant host and its root-associated bacterial communities (Table S7). We identified numerous indicators that were significantly different among the ecotypes. We further showed that, considering individual sites, there was a strong effect of the ecotypes on the indicator taxa. Our study demonstrated that plant host ecotypes matched best with their root-associated bacterial communities, showing the highest number of indicator taxa, when the plant ecotypes were at their "home field."

In the indicator taxon analysis (alpha = 0.005) across all the sites, we identified 26 indicator taxa that differed among the ecotypes. Wet ecotype was associated with the highest number of indicators ($n = 20$), followed by Mesic ($n = 6$) and Dry ($n = 0$) ecotypes (Table S8). The majority of soil bacterial indicator ASVs were assigned to Acidobacteria (total $n = 10$; Dry = 0, Mesic = 2, Wet = 8) and Proteobacteria (total $n = 10$; Dry = 0, Mesic = 1, Wet = 9). Similarly, taking into consideration the strong effects of sites, we further focused our analysis by studying the ecotypic effects at different sites. We split our data set by sites, and performed separate PERMANOVAs for each site with ecotype, population, and their interaction as the explanatory factors. We used this model separately for Carbondale, Hays, and Manhattan, to focus on the effect of ecotypes at home and away from home sites. We observed that ecotypes differed at all sites (PERMANOVA, H: $F_{2,109}$ = 2.55, $P = 0.021$; MHK: $F_{2,89} = 2.87$, $P = 0.014$; IL: $F_{2,81} = 2.55$, $P = 0.015$) (Fig. 3). In

our pairwise comparisons of community composition in Carbondale and Manhattan, wet ecotype only differed from mesic (C: Wet vs Mesic: $P_{adj}$ = 0.033, MHK: Wet vs Mesic: $P_{adj}$ = 0.042), while other ecotypic comparisons across all sites were not significant ($P_{adj}$ > 0.05).

In Carbondale, we further observed that even though the ecotypic bacterial communities dispersion was significantly different ($F_{2,79}$ = 3.84; $P$ = 0.026), the dry ecotypic communities were more dispersed than those of mesic and wet ecotypes. In addition, we did not identify ecotype-associated indicator taxa in Carbondale among the ecotypes ($n$ = 0), highlighting that the differences in ecotypic recruitment might be on a smaller scale of unique and lower abundance taxa (Table S8). In Manhattan, mesic ecotype was significantly more dispersed ($F_{2,87}$ = 10.72; $P$ = 0.01) when compared to dry and wet ecotypes. In Manhattan, we identified 11 indicator taxa: 4 indicators were associated with mesic (Actinobacteria = 2; Proteobacteria = 1; Verrucomicrobia = 2) and 7 indicators with wet ecotypes (Actinobacteria = 3; Proteobacteria = 3; Cyanobacteria = 1). We observed that in Hays, wet ecotype differed in composition from dry and mesic ecotypes (Dry vs Wet: $P$ = 0.046, Mesic vs Wet: $P$ = 0.030), but no significant differences were observed after adjusting for multiple comparisons ($P_{adj}$ = 0.05). Interestingly, ecotypes did not differ in Hays ($P_{adj}$ = 0.05). Hays is near the edge of continuous distribution of where *A. gerardii* is commonly found (37), suggesting that the drier environment might have a strong impact on diversity of bacterial populations that can proliferate under the arid conditions (94, 95). Therefore, in Hays, we hypothesize that *A. gerardii* ecotypes (i) had a lower diversity of bacterial populations for recruitment, (ii) experienced high abiotic stress from the drier conditions, and (iii) had to compete for the recruitment of the smaller microbial specialist populations. Although our results suggest that all ecotypes had a more challenging survivorship in the harsher environment, will the dry and wet *A. gerardii* ecotypes be more resilient and successful in recruiting microbes due to their "memory" from surviving in a harsh "home" environment (21, 96)? Will ecotypes get a "home-field advantage" in the recruitment of microbes at their "home" locations?

To test that, we performed the indicator taxon analysis on ecotypes at each site (Table S9). Dry ecotype recruited 64 indicator taxa ASVs in Hays, 43 in Carbondale and only 15 in Manhattan. Similarly, wet ecotype in Carbondale had the highest number of indicators (120), followed by Hays (76) and Manhattan (19). Mesic ecotype recruited 87 indicators in Carbondale, 51 in Hays, and only 7 in Manhattan. The majority of the indicator taxa from dry ecotype were Proteobacteria (41), Actinobacteria (32), followed by Acidobacteria (12). On the other hand, wet ecotype had disproportionately higher relative abundance in Proteobacteria (69) and Actinobacteria (37), followed by Verrucomicrobia (19). Even though lower microbial recruitment in Manhattan could be generally due to lower number of indicators in Manhattan as we discussed earlier, it is clear that both wet and dry ecotypes recruited more indicator taxa when they were growing in their home environments, suggesting how "home field" confer a physiological and adaptive advantage to the plant host and associated root-associated bacterial communities.

Since we observed differences in indicator taxa and unique ASVs across sites and ecotypes, we further conducted oligotyping analysis to inspect the concealed diversity within ASVs at a higher resolution (Table S10). We noticed that the relative abundance of ASVs and oligos associated with Pseudomonadales (12.9 ± 19.0%) and Rhizobiales (11.9 ± 3.6%) displayed opposite abundance patterns across the sites (Fig. 5). Pseudomonadales had lower relative abundance of ASVs across samples from Hays and Carbondale compared to samples from Manhattan (Fig. 5A). On the other hand, the relative abundance of Rhizobiales was similar throughout Hays and Carbondale and had a high degree of differences between samples in Manhattan (Fig. 5B). Interestingly, we noticed dissimilarities among sites and ecotypes across both Pseudomonadales-Oligos and Rhizobiales-Oligos (Fig. 5). Even though the relative abundance distribution patterns of oligos were obviously driven by site, we observed differences in ecotypic recruitment across ecotypes. Further work is necessary to provide more insights into recruitment

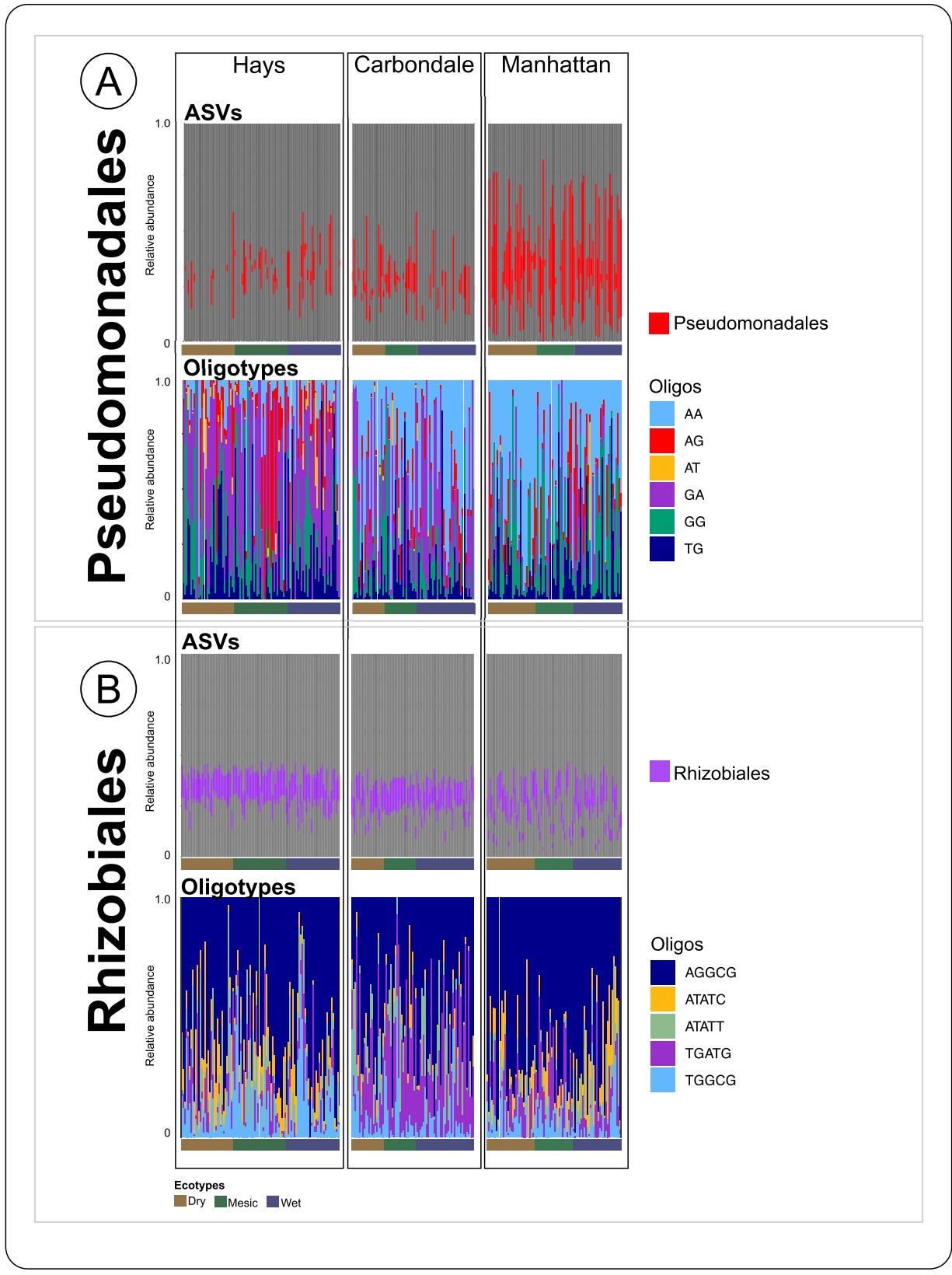

**FIG 5** Relative abundance of each oligotype within the Pseudomonadales and Rhizobiales diversity for each sample among three sites located across the precipitation gradient. The proportion of the relative abundance of the (A) Pseudomonadales (red, top) and Pseudomonadales-Oligos (bottom) and (B) Rhizobiales (purple) and Rhizobiales-Oligos (bottom) suggest the impact of locations and ecotypes on rhizobiome strain-level composition.

patterns of strain-level Pseudomonadales and Rhizobiales populations by the plant host to understand the impact of bacterial populations on hosts' resilience to environmental stresses. However, our results suggested that precipitation-induced stress (drier in Hays and wetter in Carbondale) could result in a shift of plant host–bacterial communities' interaction, changing the Pseudomonadales-Oligos and Rhizobiales-Oligos composition. For example, in Hays, Pseudomonadales-Oligos-GG were more prominent in the dry ecotype as compared to the wet ecotype, while Pseudomonadales-Oligos-AG were highly detected in mesic ecotype (Fig. 5A). On the other hand, in Manhattan, Pseudomonadales-Oligos-GA decreased in relative abundance in both dry and wet ecotypes in Manhattan, with an increase in relative abundance of Pseudomonadales-Oligos-AA. Similarly, in Carbondale, there was a higher relative abundance of Pseudomonadales-Oligos-AA in the wet ecotype as compared to dry ecotype (Fig. 5A). We further noticed that in Hays, there was a higher relative abundance of Rhizobiales-Oligos-ATATC in dry ecotype as compared to wet ecotype. However, the relative abundance of Rhizobiales-Oligos-ATATC was higher in the wet ecotype as compared to dry ecotype in Manhattan. Interestingly, Rhizobiales-Oligos-TGATG was substantially higher in relative abundance in wet and mesic ecotypes as compared to dry ecotype in Carbondale (Fig. 5B). The differences in the Pseudomonadales-Oligos and Rhizobiales-Oligos across the sites potentially highlight the tendency of plants to recruit microbial strains that could contribute to the plant hosts' ecotypic resilience (21, 97). The intraspecific genome content may be particularly important in understanding these host–microbe interactions. The intraspecific genome content may be particularly important in understanding these host–microbe interactions.

## Conclusion

Plant responses and adaptation to stress depend on a combination of environmental factors and plant genetics (1, 2). While the environment defines the soil-associated microbial pool, our results highlighted the plant-host's ability to recruit soil microbial communities under different environmental conditions. We showed that plant hosts were more successful in the recruitment of both the general and unique microbial populations when grown at their homesite (98). Our observations also suggested the "home-field advantage" and mutual association between plant and specific soil microbes (21, 22, 97). From our study, we further propose that, at their home sites, ecotypes originating from these harsher environments and experiencing constant abiotic stresses were better able to recruit specialist microbes with the potential stress relief functions (98). We argue that in the study of plant host and root-associated bacterial communities, in line with the generalist and specialist hypothesis, plant host's resiliency under abiotic stress is more dependent on specialized microbes (2, 4, 73, 86, 87) rather than the core microbiome (15, 99–101). Our study suggests that in terms of the resistance to abiotic stress, the key factor is less abundant and specialized microbes with specific functions rather than general common microbes which are abundant in soil. We further observed, in our study, the fragile relationships between plant hosts and associated specialized microbes. Although it might be challenging to tease apart the "specialist vs generalists" and "home-field advantage" concepts, we used oligotyping analysis and showed the ecotypic variations in recruitment of bacterial stains from the same genera. These observations further highlighted the complexity of these ecotype–host relationships. After growing in a common garden for over 10 years, we could still see the distinct microbial communities recruited by the ecotypes. Therefore, these differences across rhizosphere microbiomes recruited by ecotypes across all sites demonstrated that the plant host–bacterial interaction is much more exclusive and fragile than we imagined. We believe that ecotypes rely on these relationships with microbes to overcome abiotic stress (102), and with the change in climate, these relationships might be threatened. Taking all these factors into consideration, there is a crucial gap in identifying specialist microbes and understanding these relationships and communication signals with plant hosts to predict the response of species to the changing climate change.

## ACKNOWLEDGMENTS

This work was supported by the National Science Foundation EPSCoR Award No. OIA-1656006 and matching support from the State of Kansas Board of Regents. This study was supported by the United States Department of Agriculture, National Institute of Food and Agriculture (USDA NIFA), under the Award Number: 2020-67019-3180.

We would like to thank Sara Baer for aiding in the designing of the reciprocal gardens. We would like to thank all of those who assisted in sampling and data acquisitions. We are grateful for the help of Tanner Richie, Brandi Feehan with the discussion on bioinformatics and data analysis. We are grateful for the technical assistance by Alina Akhunova, Sarah Bastian, and Samantha Elledge at the Integrated Genomic Facility at Kansas State University (https://www.k-state.edu/igenomics/). We thank Sara Baer (Kansas Biological Survey & Center for Ecological Research) and the Kansas State University Soil Testing Laboratory for helping with the soil chemistry and parameter analyses.

A.J., L.J., and S.T.M.L. designed the study. L.J. designed the reciprocal gardens experimental design, and A.J., L.J., and S.T.M.L. maintained the sites. Sample collection was performed by S.S., E.H., M.G., A.J., and S.T.M.L., whereas S.T., S.S., A.K., and K.W. extracted DNA with Nanodrop and Qubit quality analysis. A.K. and Q.R. performed 16S rRNA sequencing bioinformatic analysis. Q.R. and S.T.M.L generated oligotypes data. A.K. performed statistical analyses. A.K. and S.T.M.L wrote the manuscript, prepared figures, and supplementary files. A.J. and L.J. aided in manuscript and figure revisions. S.T.M.L., A.J., and L.J. acquired funding for this study. All authors read, contributed to manuscript revision, and approved the submitted version.

The authors disclose no conflicts of interest.

## AUTHOR AFFILIATIONS

[1]Division of Biology, Kansas State University, Manhattan, Kansas, USA
[2]Department of Biology, University of North Carolina, Greensboro, North Carolina, USA
[3]Department of Biological Sciences, Fort Hays State University, Hays, Kansas, USA

## AUTHOR ORCIDs

Anna Kazarina  http://orcid.org/0000-0001-8782-0834

## FUNDING

| Funder | Grant(s) | Author(s) |
| --- | --- | --- |
| National Science Foundation (NSF) | EPSCoR Award No. OIA-1656006 | Ari Jumpponen |
| | | Loretta Johnson |
| | | Sonny T. M. Lee |
| USDA \| National Institute of Food and Agriculture (NIFA) | 2020-67019-3180 | Sonny T. M. Lee |

## AUTHOR CONTRIBUTIONS

Anna Kazarina, Conceptualization, Data curation, Formal analysis, Investigation, Methodology, Software, Supervision, Validation, Visualization, Writing – original draft, Writing – review and editing | Soumyadev Sarkar, Funding acquisition, Project administration, Validation, Visualization, Writing – review and editing | Shiva Thapa, Data curation, Validation, Visualization, Writing – review and editing | Leah Heeren, Data curation, Visualization, Writing – review and editing | Abgail Kamke, Data curation, Visualization, Writing – review and editing | Kaitlyn Ward, Data curation, Visualization, Writing – review and editing | Eli Hartung, Data curation, Formal analysis, Visualization, Writing – review and editing | Qinghong Ran, Data curation, Formal analysis,

Software, Supervision, Visualization, Writing – original draft, Writing – review and editing | Matthew Galliart, Data curation, Formal analysis, Validation, Writing – review and editing | Ari Jumpponen, Conceptualization, Funding acquisition, Project administration, Resources, Writing – review and editing | Loretta Johnson, Conceptualization, Funding acquisition, Project administration, Resources, Writing – review and editing | Sonny T. M. Lee, Conceptualization, Data curation, Formal analysis, Funding acquisition, Investigation, Methodology, Project administration, Resources, Software, Supervision, Validation, Visualization, Writing – original draft, Writing – review and editing

## DATA AVAILABILITY

The raw sequence data are available through the Sequence Read Archive under BioProjectBioSamples PRJNA911775, while bioinformatic and statistical codes are available through figshare.

## ADDITIONAL FILES

The following material is available online.

### Supplemental Material

**Table S1 (Spectrum00208-23-S0001.xlsx).** Locations of collected seeds.
**Table S2 (Spectrum00208-23-S0002.xlsx).** Samples.
**Table S3 (Spectrum00208-23-S0003.xlsx).** ASV assignments.
**Table S4 (Spectrum00208-23-S0004.xlsx).** Soil chemistry.
**Table S5 (Spectrum00208-23-S0005.xlsx).** Indicator taxon locations.
**Table S6 (Spectrum00208-23-S0006.xlsx).** Venn diagrams.
**Table S7 (Spectrum00208-23-S0007.xlsx).** Indicator taxon ecotypes.
**Table S8 (Spectrum00208-23-S0008.xlsx).** Indicator taxon individual locations.
**Table S9 (Spectrum00208-23-S0009.xlsx).** Indicator taxon ecotypes per location.
**Table S10 (Spectrum00208-23-S00010.xlsx).** Genus-level oligotyping analysis.

### Open Peer Review

**PEER REVIEW HISTORY (review-history.pdf).** An accounting of the reviewer comments and feedback.

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
