## [Reviewer comments · Microbiology Spectrum]

Microbiology Spectrum

Home-field advantage affects the local adaptive interaction between *Andropogon gerardii* ecotypes and root-associated bacterial communities

Anna Kazarina, Soumyadev Sarkar, Shiva Thapa, Leah Heeren, Abigail Kamke, Kaitlyn Ward, Eli Hartung, Qinghong Ran, Matthew Galliard, Ari Jumpponen, Loretta Johnson, and Sonny Lee

Corresponding Author(s): Sonny Lee, Kansas State University

Review Timeline:

Submission Date:	January 14, 2023
Editorial Decision:	February 19, 2023
Revision Received:	April 27, 2023
Editorial Decision:	June 9, 2023
Revision Received:	June 28, 2023
Accepted:	July 5, 2023

Editor: Renee Arias

Reviewer(s): The reviewers have opted to remain anonymous.

Transaction Report:

DOI: <https://doi.org/10.1128/spectrum.00208-23>

February 19, 2023

Dr. Sonny T.M. Lee
Kansas State University
Division of Biology
Manhattan

Re: Spectrum00208-23 (Home-field advantage affects the local adaptive interaction between *Andropogon gerardii* ecotypes and rhizobiome)

Dear Dr. Sonny T.M. Lee:

The author would need to carefully address all the comments from the reviewers.

Thank you for submitting your manuscript to Microbiology Spectrum. A number of concerns have been listed by the reviewers; these will need to be carefully addressed. When submitting the revised version of your paper, please provide (1) point-by-point responses to the issues raised by the reviewers as file type "Response to Reviewers," not in your cover letter, and (2) a PDF file that indicates the changes from the original submission (by highlighting or underlining the changes) as file type "Marked Up Manuscript - For Review Only". Please use this link to submit your revised manuscript - we strongly recommend that you submit your paper within the next 60 days or reach out to me. Detailed instructions on submitting your revised paper are below.

Link Not Available

Sincerely,

Renee Arias

Journals Department
Reviewer comments:

Reviewer #1 (Comments for the Author):

This research seeks to determine the effect of ecotype and location in big bluestem ecotypes planted across sites running from west to east exhibiting a substantial precipitation differences. Ecotype seeds representing four populations were recovered previously from each of the three sites inclusive of the genotypic adaptation to local environmental conditions. Big Bluestem plants were established first in a greenhouse and then transplanted into the what is termed a reciprocal garden where each ecotype, population is represented at each of three locations. Plants were then sampled using a soil core technique and DNA extracted and analyzed using a number of statistical routines. The project seeks examine whither microbiome assembly is associated with a "home field advantage" whereby soil bacteria from a given location is more apt to colonize plants adapted to that location.

This is a very interesting study in a number of ways- 1 the target being big bluestem a major prairie species in the great plains grass ecosystem that is under studied in terms of rhizobiome development. 2- the use of reciprocal plantings of local and non-local ecotypes to tease out the effect of environment and genotype on microbial assembly, 3 the use of minimum entropy decomposition in conjunction with an oligo typing methodology to identify sequence variants in comparison to transcript alignment procedures 4- the use of three sites widely separated and differing in precipitation characteristics.

The research as outlined clearly had a number of interesting facets as indicated above. Overall the research clearly shows the predominance of location in the formation of the rhizobiomes which appears to swamp the ecotype effect.

However as written the paper suffers a number of aspects that needs addressing before a clear understanding of the work can be obtained.

L141. How were the populations different genotypically. It appears from the reference, that the seeds were obtained on site at each location and referred to as populations from each of the three locations. How different were these populations from each other and especially among ecotypes? This would be important background information. Currently we have no idea about the genotypic differences associated with the treatment experimental design.

L148 The seeds from each population were first germinated in potting soil mix in a greenhouse environment. Then the seedlings were transported to the field. Why were they not established directly in the field? Could the establishment method have introduced an artificial aspect into the rhizobiome prior to sampling, given their perennial nature? How long were the plants established before sampling took place? This is not indicated at all. If everything occurred in the first year then there may be a problem.

L 154

1) Also the work represents only a single years of data. In almost all field studies multiple years are required to come to a certain conclusion.

2) At what time of year were the plants sampled. What were the conditions at and prior to sampling in terms of soil moisture and other factors for each site? Especially for soil moisture.

3) What are the characteristics of the soils at each location is left completely unstated in terms of soil type, soil structure, and soil nutrients. These may present alternative explanations associated with the findings.

4) how was the sampling performed beside just coring the soil. What location in relationship to the plant root system? Was a single core obtained from each plant or multiple cores. Were these cores bulked and then subsampled? Single cores would increase experimental variability, bulking would produce a more representative sample. Also, the 15 cm cores are likely only to contain soil from surface roots, but given the deep rooting nature of big bluestem does such a core really represent the plants rhizobiome?

L155 Some plants died during establishment. How did this affect the sampling distribution among sample cores? Did any ecotype population suffer from this dieback? This needs to be stated

L160 Did the authors extract DNA and sequence all 284 cores? How does the sampling strategy feed into the DNA sequencing?

L164 Since the authors extracted the roots from the soil cores and shook off the non rhizosphere soils and then extracted the DNA then the results represent not only the rhizosphere but also the endorhizosphere organisms. This should be stated.

L200 The authors performed indicator analysis using a wide range of computer algorithms, but it is unclear exactly how this was done. More detail would be appreciated.

L217 The authors performed analysis on 110 samples. (2,104,416/19,135) How does this relate to the 284 plants. Again there is confusion concerning the sampling system? See L160 comment

L233 The table only presents p values and not absolute values for each of the indices so we really can't judge whether one indices is higher or not.

L234 The data only supports a lower diversity part of the definition.

L236-237, 248 intermediate conditions in terms of what factor? There are likely multiple factors working on this system and the authors only focus on moisture. That is not very convincing.

L251 pH is certainly a factor that affects bacterial diversity in soils. The authors do not present any evidence that pH differed in the soils. The idea presented here is that drier soil due to lower moisture increases pH. There are other factors that are more important than moisture that determines soil pH such as soil substrate. Soil data should be presented

L 263 were the soil waterlogged at time of sampling. No data presented.

L308 The numbers in the Venn diagram are used to indicate the home field advantage. It would be much better if there were statistics attached to these numbers to determine if there was a statistical difference. As is, there is only one replication which does not lend support for the contention for the most important objective of this work.

L328 The authors state that increased photosynthetic rates under dry conditions would favor microbial specialists. Again this is sheer speculation without any supporting data, only a reference. The sentence should be phrased as such.

L358 Why are there no indicator species in the dry habitat?

L392 This data since it is discussed should be presented as a figure or table rather than relegated to the supplement.

Table 1 is out of focus. Should be a sharper image. Also it presents only p values. The table could be reconstructed to present both absolute values as well as P values.

Comments are included in the body of the manuscript

Reviewer #2 (Comments for the Author):

In their article, 'Home-field advantage affects the local adaptive interaction between *Andropogon gerardii* ecotypes and rhizobiome' the authors used amplicon metabarcoding of bacterial ribosomal genes to detect local adaptation in rhizosphere associated bacterial communities in a common garden experiment. The authors demonstrated that site was the major driver of bacterial diversity indexes and community composition, with ecotype also contributing to the presence of unique taxa. The design of the common garden experiment, paired with the amplicon metabarcoding, could allow the evaluation of local adaptation to environments with different soil moisture and precipitation. However, the statistical analysis, as presented does not test for interaction effects between site and ecotype. Therefore, the results and discussion are presented either for site or for ecotype, but no interaction between them. Further the results and discussion are difficult to follow, with analyses repeated across sections.

The manuscript presents information on environmental characteristics at each site, such as average precipitation and temperature. However, I think for this type of study that evaluates differences in soil moisture across sites, it will be important to include information on the characteristics of the soil at the time of sampling (for example, time from the last rain event, or soil moisture/temperature at the time of sampling), as rain events could cause shifts in the communities of bacteria in the sample. This could then influence the results and their interpretation.

As this article focuses on bacterial communities, I recommend the use of language that highlight this throughout the manuscript. For example, the use of terms, such as bacterial instead of microbial, or bacterial communities instead of microbiome. This is important as the responses to environmental stressors or local adaptation between groups of microbes (e.g. bacteria vs. fungi) could differ.

Additional comments are listed below by section.

Abstract

Ln41. I am not sure I understand the sentence, if possible, rewrite '...data also suggest that ecotypes were more successful in recruiting rhizosphere community members unique to their local homesites'. Does this mean that when the ecotype, whose seed was collected on a mesic site, was planted at the mesic site, then a greater number of bacterial taxa unique to the ecotype and site combination were recovered?

Ln 44. The evidence supports that unique microbes could be related to adaptation to sites with variations in water availability. No data is presented regarding plant stress responses. I recommend removing the statement regarding plant stress responses.

Ln 48-50. The last two sentences of the abstract could be rewritten for clarity. For example, Ln 61-62 is a similar statement and easier to understand.

Introduction

Ln 92. Given the emphasis on "home field advantage" in this manuscript, I recommend including definitions and examples of these.

Ln 119-122. Are (1) and (2) in this paragraph different objectives? It seems that (1) states the objective and (2) the approach. In this work, the authors investigated the "relative importance of the plant local environment and phenotypic variation ...on establishing plant rhizobiome" by comparing rhizosphere bacterial communities "under reciprocally transplanted, local and non-local adapted environments".

Ln 122. Number (3) indicates examining potential links between members of the bacterial community and plant hosts' resilience characteristics. There is no evidence presented in this article that relates to specific plant traits that could contribute to adaptation to different soil moisture environments, nor analysis performed to determine the relationship between specific plant traits and bacterial communities. In different places in the manuscript, prior studies are referenced, which do summarize specific traits identified in individual ecotypes. But the current analysis focus on ecotypes and the location of common gardens.

Materials and Methods:

Ln 178-179. By runs, do you mean sequencing runs? If so, how many sequencing runs were needed to process the analyzed samples? Also, what were the parameters used for the DADA2 pipeline. Provide a similar level of detail as it was provided for the section on primer trimming.

Ln. 179. What is the justification for rarefying the data set to 10,000 sequences per sample (and not 20,000 or 50,000 sequences per sample, for example)? By briefly looking at the supplementary tables, it seems a higher number could have been used and still not lose too many samples with low reads. Also, is it appropriate to assume that rare taxa (which will be lost by rarefying to 10,000) have less relevance for studies on local adaptation or home advantage? Will higher sequencing depth per sample result in similar results?

Ln 184-202. Indicate which portions of the analysis were performed in Qiime2, and which in R. For specific analysis include the

functions used, and when appropriate indicate references. For example, is the betadispr function a function from which package, and is there a reference that can be used to replicate these analyses?

Ln 213. What were the relative abundances of *Pseudomonas* and *Rhizobium*? Also, why only high abundance taxa were selected? Could it be possible that less abundant or rare taxa be relevant for this study? Finally, in this section the genus name is used, however, the discussion and figures are at the order level (*Pseudomonadales* and *Rhizobiales*), not the genus level (*Pseudomonas* and *Rhizobium*). For consistency, I recommend using the order in this section as well.

Results and discussion.

Be consistent in the use of the terms site or location, but not the combination of 'site locations'. Also, be consistent in the use of these terms in the figures and legends. For example, Figure 1 legend uses the word site in Ln 723, then location in 725, 'site location' in 727, and location in the figure key. In general, this section is difficult to read, with results of different analyses (ANOVA, PERMANOVA, unique taxa, and indicator taxa) presented in different sections, and not a clear synthesis.

Ln 219 - Add a reference for the classifier tool used

Ln 220 - Were plant reads (chloroplast or mitochondria) also removed from analyses?

Ln 227 - What were the results of the full test, including both main effects and interactions? The materials and methods (Ln 184) state that the analysis was used to test the main and interactive effects in the various diversity indexes tested. Were there significant effects for the interactions between location, ecotype, and population?

Ln 234 - How did you determine that a location was more favorable for generalist, fast-growing microbes? It is not clear how phylogenetic diversity data can be used to infer this.

Ln 240-246. Did the PERMANOVA model include location*ecotype interaction?

Ln 250-254. In addition to precipitation gradient and pH are there other environmental parameters that vary across sites that could influence the results?

Ln 294. Why is a different ANOVA run? Were the ecotype and populations not included in the initial analysis?

Ln 308. Do unique taxa refer to taxa that are only present on an ecotype regardless of the population?

Ln 307-322. It is not clear how the bacterial taxa are considered either generalist or specialist, or "good plant colonizers"

Ln 320-321. Did the referenced studies (63 and 36) document adaptation to environments with different water availability, or studies that focus on tolerance to drought stress? Do these report quantitative differences in traits in the different environments?

Ln 321. Include references that support the statement that lower photosynthetic rate results in lower amounts of root exudates or other deposits that could influence microbial recruitment in the root.

Ln 349-354. Should this paragraph be in the previous section? Both the analysis of unique taxa and indicator species contribute to identifying taxa that could be associated with either (or both) location and ecotype.

Ln 356-366. Are the PERMANOVA results here different from the PERMANOVA results discussed in the prior section (Ln 303)?

Ln 406-409. The section begins with *Pseudomonas* and *Rhizobium*, and continues the discussion at the order level. Be consistent in the taxonomic level used for analysis and discussion.

Ln 404-434. This section focuses on the analysis of specific oligos or sequence types belonging to the *Pseudomonadales* and *Rhizobiales*. Are there statistical tests that can be applied to these results to provide support to the statements based on the visualization of sequence type distribution across locations and ecotypes (and their interaction)?

Conclusion

Ln 457. What do you mean by 'the disconnect' between the plant host ecotype and rhizobiome?

Staff Comments:

Preparing Revision Guidelines

Please return the manuscript within 60 days; if you cannot complete the modification within this time period, please contact me. If you do not wish to modify the manuscript and prefer to submit it to another journal, please notify me of your decision immediately so that the manuscript may be formally withdrawn from consideration by Microbiology Spectrum.

**Home-field advantage affects the local adaptive interaction**
**between *Andropogon gerardii* ecotypes and rhizobiome**

Anna Kazarina¹, Soumyadev Sarkar¹, Shiva Thapa², Leah Heeren¹, Abigail Kamke¹, Kaitlyn
Ward¹, Eli Hartung¹, Qinghong Ran¹, Matthew Galliard³, Ari Jumpponen¹, Loretta Johnson¹, Sonny
6 T.M. Lee^{1*}

¹ Division of Biology, Kansas State University, Manhattan, Kansas, United States

² Department of Biology, University of North Carolina at Greensboro, North Carolina, United
States

³ Department of Biological Sciences, Fort Hays State University, Hays, Kansas, United States

* Corresponding author, e-mail: leet1@ksu.edu

***Keywords:** Rhizosphere, plant microbiome, ecotypic variation, local adaptation, precipitation
gradient

**Abstract**

Due to climate change, drought frequencies and severities are predicted to increase across the
United States. Plant responses and adaptation to stresses depend on plant genetic and
environmental factors. Understanding the effect of those factors on plant performance is required
to predict the species responses to environmental change. We used reciprocal gardens planted
with distinct regional *Andropogon gerardii* ecotypes adapted to dry, mesic and wet environments
to characterize their rhizosphere communities using 16S rRNA metabarcoding sequencing. Even
though the local microbial pool was the main driver of these rhizosphere communities, the
significant plant ecotype effect highlighted active microbial recruitment in the rhizosphere driven
by ecotype or plant genetic background. Our data also suggest that ecotypes were more
successful in recruiting rhizosphere community members unique to their local homesites,
supporting the “home field advantage” hypothesis. These unique homesite microbes may
represent microbial specialists that are linked to plant stress responses. Further, our data support
ecotypic variation in the recruitment of congeneric but distinct bacterial variants, highlighting the
nuanced effects of plant ecotypes on the rhizosphere microbiome recruitment. Our results should
facilitate expanded studies on understanding the complexity of plant host interactions with local
soil microbes and identification of functional potential of recruited microbes. Our study has the
potential to aid in predicting ecosystem responses to climate change and the impact of
management on restoration practices.

**Importance**

In this study, we used reciprocal gardens located across a sharp precipitation gradient to
characterize rhizosphere communities of distinct dry, mesic, and wet regional *Andropogon*
*gerardii* ecotypes. We used 16S rRNA amplicon sequencing and focused oligotyping analysis and
showed that even though the location was the main driver of the microbial communities, ecotypes
could potentially recruit distinct bacterial populations. We showed that different *A. gerardii*
ecotypes were more successful in overall community recruitment and recruitment of microbes
unique to the “home” environment, when growing at their “home site”. We found evidence for
“home field advantage” interactions between the host and associated rhizobiomes, and the
capability of ecotypes to recruit specialized microbes that were potentially linked to plant stress
responses. Our study provides insights into the understanding of factors effecting the plant
adaptation, improving management strategies, and predicting of the future landscape under the
changing climate.

Introduction

The rhizosphere is a dynamic region characterized by the complex interactions between the plant
host and associated microbial communities [1,2]. It has been widely recognized that plants directly
and indirectly benefit from associated microbial activities and resultant microbial compounds [3,4].
Complex microbe-microbe interactions around the rhizosphere result in facilitating nutrient
transformations, uptake, and cycling as well as altering soil structure and soil water availability
[5,6]. Similarly, plant host-microbe interactions can also greatly affect the overall plant health and
productivity [7,8]. Root-associated microorganisms (hereafter referred to as rhizobiome) can
affect plant resistance to biotic and abiotic stress, help with nutrient uptake and even alter plant
morphology and phenology [9,10]. Given the crucial roles of the rhizobiome, understanding what
shapes microbial community assembly, function, and mechanisms, as well as adaptive responses
between plant host and associated microbes is critical in predicting the response of this system
to changing environmental conditions [11,12].

Numerous studies have demonstrated the need to consider the interactive effects of the
environment and plant host genetics in understanding the rhizobiome. The chemical and physical
characteristics of the local soil can directly affect plant function, which in turn can have
consequential influence on the rhizobiome composition [11,13,14]. Plant hosts actively modulate
associated microbial communities by releasing various signaling molecules (phytohormones) and
compounds into the soil [11,15]. Phytohormones are structurally diverse secondary metabolites
released by the plant to activate the immune system [16,17] in response to microbial pathogens
[18] and even insect herbivores [19]. In addition to direct immune responses, plants produce
phytohormones and compounds in response to abiotic stress such as nutrient or water deficiency,
or to promote symbiotic interactions with soil microbes [20]. Although there are numerous studies
on plant host influence on rhizobiome composition, only little information is available about the
interactive impact of the host-environment on the rhizobiome. The rhizobiome is a subset of
microorganisms available in the surrounding soil microbial pool [15]. Thus, it is no surprise that
the same plants perform differently in distinct locations. Some studies have described an increase
in efficacy of the plant host-associated rhizobiome interaction due to the “home field advantage”
[21,22]. However, questions remain if plants locally adapted to the prevailing environmental
conditions can take advantage of the local soil microbes or if plants preferably favor microbiomes
similar to their home environment.

Due to climate change, drought frequencies and severity are predicted to increase across the
United States [23-25]. In the Great Plains, drought events limit productivity especially in tallgrass
prairies [24]. Thus, understanding the effect of drought in shaping the rhizobiome and the
mechanisms of mutual adaptation between plants and associated microbes is critical in the
prediction of the ecosystem's response to changing climate. To date, most studies [26-28] on the
effect of drought on plant-microbe interactions have been under *in vitro* conditions using model
organisms, lacking the complexity and dynamics of the natural system application [29,30].

To address the question if there is a co-adaptation of the plant host and its associated rhizobiome
in a non-model plant system, our study took the opportunity to investigate the rhizobiome
composition of three locally adapted big bluestem (*Andropogon gerardii* Vitman) ecotypes planted
in reciprocal common gardens across the precipitation gradient in the Great Plains. *Andropogon*
*gerardii* is a perennial C4 grass that is widely distributed across the Great Plains of North America
[31,32], and covers up to 80% of the biomass in tallgrass prairie [32-35]. Within the Great Plains,
*A. gerardii* has been growing for over 10,000 years along prominent sharp rainfall gradients that
range from semiarid to heavy rainfall [36,37]. Time and environmental heterogeneity lent support
for local adaptation of *A. gerardii*, giving rise to distinct ecotypes (dry, mesic, and wet) [36,38,39]
[36,39,40]. Previous investigations have revealed that *A. gerardii* ecotypes vary in functional traits
that influence microbially mediated processes [40,41]. Although there are studies investigating
intraspecific [42] and interspecific (reviewed in [40]) plant responses to climate [43], the role of
the rhizobiome in plant host adaptation in the natural system remains unclear.

Here, we (1) investigated the relative importance of the plant local environment and phenotypic
variation of *A. gerardii* on establishing plant rhizobiome; (2) compared ecotypic responses in their
ability to recruit rhizosphere microbes under reciprocally transplanted, local and non-local
environments; and (3) examined rhizosphere community members with the potential to link the
microorganism recruitment and the plant hosts' ecotypic resilience functions. We hypothesized
that *A. gerardii* ecotypes would perform better at the location closely matching their "home"
environment, highlighting the effect of the plant genetic background or ecotype on rhizosphere
community assembly. We also predicted that greater diversity or locally adapted unique microbes
would be recruited in the homesites. Our work contributes to a clearer understanding of the factors
that influence the plant host's recruitment of rhizobiome, and will help to provide insights into *A.*
*gerardii* ecotypic responses to a future changing climate.

**Materials and Methods**

***Study sites, samples collection and processing***

We sampled three *A. gerardii* reciprocal gardens in the summer of 2019. The gardens had been
established in 2009 and continually maintained at the three sites: Hays, KS (H) at Kansas State
University Agriculture Experimental Station (38°85'N, 99°34'W), Manhattan, KS (MHK) at USDA
Plant Materials Facility (39°19'N, 96°58'W), and Carbondale, Illinois (C) at Southern Illinois
University Agriculture Research Station (37°73'N, 89°17'W), giving us an excellent opportunity to
study the interactive effects of local environment and hosts on the rhizobiome (Supplementary
Table S1). Experimental details of the reciprocal garden experiment have been published in
Galliard et al. [44]. Briefly, in 2009, seeds were collected from four *A. gerardii* populations at each
of three different locations (12 populations in total): mixed grass Central Kansas (CKS/Dry: CDB,
REL, SAL, WEB), tallgrass Eastern Kansas (EKS/Mesic: CAR, KON, TAL, TOW) and Southern
Illinois savanna (SIL/Wet: 12MI, DES, FUL, WAL) (Acronyms of populations are listed in
Supplementary Table S1) prairies across the natural rainfall gradient with 580 mm/year, 871
145 mm/year and 1167 mm/year of mean annual precipitation, respectively. We defined the ecotypic
variation of *A. gerardii* based on the locations they were collected (Dry: Hays; Mesic: Manhattan
and Wet: Carbondale). Seeds representing the three ecotypes and twelve populations were
germinated and grown in a greenhouse using potting mix (Metro-Mix 510). Established 3- to 4-
149 months-old seedlings were then planted at each three reciprocal garden sites (size - 4 × 8 m), in
which 12 plants (4 populations x 3 ecotypes) were planted in a complete randomized block design
with 10 blocks (rows) for a total of 120 plants per site. Plants were planted 0.5 m apart along each
row, and the soil around the plants was covered with the water-penetrable landscape cloth for
weed control.

Some plants did not survive transplanting or through the ten years of growth in the common
gardens. As a result, we were able to sample a total of 284 *A. gerardii* rhizospheres across the
three locations (Supplementary Table S2) using a soil core (15 cm deep x 1.25 cm diameter).
Samples were sealed in ziplock bags, transported on ice, and stored at -20°C until DNA
extraction.

We extracted total DNA from 0.150 g of rhizospheric roots and soil using an Omega E.Z.N.A. Soil
DNA Kit (Omega Bio-Tek, Inc., Norcross, GA, USA) as per the manufacturer's protocol with a
slight modification. We removed soil from the plant roots by shaking; any soil that remained

attached to the roots was considered rhizosphere soil [45,46]. We mechanically lysed the cells on
a Qiagen TissueLyser II (Qiagen, Hilden, Germany) using glass beads for 2 mins at 20 rev/s prior
to any downstream DNA extraction steps. The extracted DNA was eluted to 100 μ L final volume.
The DNA yield and concentration was measured using a Nanodrop and a Qubit™ dsDNA BR
Assay Kit. Extracted DNA was sequenced (2 x 250 cycles) using Illumina MiSeq with the 16S
rRNA V4 region amplified using the primers 515F and 806R at the Kansas State University
Integrated Genomics Facility.

**Sequence data processing and analyses**

We used QIIME 2 v. 2021.4 [47] to process a total of 8,353,179 raw sequences, resulting in
5,628,314 bacterial sequences after quality control. We used QIIME 2 plugin cutadapt [48] to
remove the primer sequences. Any sequences with ambiguous bases, with no primer, with greater
than 0.1 error rate mismatch with primer or any mismatches to the sample-specific 12 bp
molecular identifiers (MIDs) were discarded. Following initial quality control, we used DADA2 [49]
with the same parameters across different runs. We used the pre-trained SILVA database (v. 138)
in QIIME 2 for taxonomic assignment of the bacteria. We rarefied the data set to 10,000 reads
180 per sample (resulting in 2,104,416 high quality sequences) to minimize biases resulting from
181 differences in sequencing depth among samples before estimating diversity indices and
182 downstream analyses [50].

We used Analysis of Variance (ANOVA) to test for main and interactive effects in observed
richness (S_{Obs}), community (Shannon's H'), and phylogenetic (Faith's PD) diversity of the
rhizosphere associated bacterial communities among the locations (Hays (H), Carbondale (C),
Manhattan (MHK)), ecotypes (dry, mesic, wet), and populations (12MI, CAR, CDB, DES, FUL,
KON, REL, SAL, TAL, TOW, WAL, WEB) nested within ecotype. Following overall ANOVA, we
used pairwise comparisons using Kruskal-Wallis test to identify factors that were driving the
significant effects.

We estimated pairwise Bray-Curtis distances to compare the bacterial communities among the
different factors, and visualized these data using Non-Metric Multidimensional Scaling (NMDS)
ordinations. We used a nonparametric permutational analog of traditional analysis of variance
(PERMANOVA) to determine whether bacterial composition differed among locations, ecotypes
and populations. In this model, "population" was nested within "ecotype." Following the
PERMANOVA, we performed pairwise comparisons using pairwise.adonis function. We also used

betadispr function to test whether the community dispersion differed between any significant
groups. To identify bacterial populations that were disproportionately more abundant in one
significant group than in another, we analyzed these data using indicator taxon analyses. The
community composition data analyses were conducted in R ape [51], indicpecies [52], lme4 [53],
phyloseq v 1.42.0 [54], stat v 4.1.1, vegan [55], R studio [56].

**Oligotyping analyses**

We used minimum entropy decomposition (MED) [57] algorithm with default parameters to identify
sequence variants in high-throughput 16S rRNA amplicon sequences. The MED partitions the
sequences into discrete sequence groups by minimizing the total entropy in the dataset. We then
concatenated the sequences assigned to a specific genus using silva 138 reference database
and used the supervised oligotyping method available in the oligotyping pipeline version 2.1 [58].
In this supervised oligotyping method, we used Shannon entropy, with a threshold value of
minimum 0.2 and minimum substantive abundance threshold of 10, to obtain genus-level
oligotypes. We selected *Pseudomonas* and *Rhizobium* oligotypes for further analysis due to the
overall high relative abundance across the bacterial ASVs and generated oligotypes (Oligos:
*Pseudomonas*-Oligos (n=6); *Rhizobium*-Oligos (n=6))

**Results and discussion**

We analyzed a total of 2,104,416 (19,131 ± 4,115 per sample) high quality sequences assigned
to 1,157 ASVs. Of the recovered reads, an average of 90% was annotated to the genus level on
Naive Bayes classifier. Any unknown or unclassified ASVs were removed from downstream
analyses. The bacterial taxon assignments along with their counts in each sample are provided
in Supplementary Table S3. Our data were dominated by the phylum Proteobacteria (45% of all
sequences), followed by Actinobacteria (15%), Acidobacteria (12%) and Bacteroidata (6%). A
small proportion of sequences remained unclassified (0.29%). ASV relative abundances were
dominated by Proteobacteria (28% of all ASVs), Actinobacteria (14%), Firmicutes (12%) and
Bacteroidetes (9%). Similar to sequences, the proportion of unassigned ASVs was small (0.3%).

**Location had a strong impact on the ecotypic rhizobiome richness and diversity**

Location was the main driver of the bacterial communities, with a strong effect on bacterial
richness and diversity (ANOVA, S_{Obs} : $F_{2,245} = 7.13$; $P < 0.001$; Shannon's H' index: $F_{2,245} = 26.56$;
$P < 0.001$; Faith's PD index: $F_{2,245} = 4.09$; $P = 0.018$) (Figure 1A). Pairwise tests showed that
community richness (S_{Obs}) and Shannon's diversity were lower in Manhattan than in the other two

locations (Table 1). On the other hand, the phylogenetic diversity (Faith's PD) in Carbondale was
marginally higher than in Hays and Manhattan (Table 1). Phylogenetic diversity analyses suggest
that Manhattan was favorable for generalist microbes - fast growing and abundant soil microbes
that are good at colonizing the plant and often characterized by the lower diversity [2,4] due to
the intermediate environmental conditions as compared to Hays and Carbondale. The higher
bacterial phylogenetic diversity we observed in this study could be highly attributed to the amount
of precipitation among the locations.

Similar to the alpha-diversity, the bacterial communities clustered by location indicate that
locations had strong effects on bacterial composition (PERMANOVA, $R^2 = 0.42$; $F_{2,245} = 102.53$;
$P = 0.001$; Figure 1B). Our pairwise comparisons further corroborated that all locations had
significantly different bacterial compositions (Pairwise Adonis, C vs H: $R^2 = 0.435$; $P = 0.001$; P_{adj}
$= 0.003$; C vs MHK: $R^2 = 0.313$; $P = 0.001$; $P_{adj} = 0.003$; H vs MHK: $R^2 = 0.274$; $P = 0.001$; $P_{adj} =$
0.003). We conducted a dispersion analysis, and observed that bacterial communities in
Manhattan ($F_{2,279} = 3.899$; $P = 0.015$) were more varied than Carbondale ($P = 0.020$). Similar to
richness, the variation in bacterial community compositions among the locations were likely a
result of precipitation-associated biotic and abiotic differences [59] as well as interspecific
microbial competition [2,4]. For example, the mean precipitation in Carbondale is 1167 mm/yr
which is drastically different from that in Hays (580 mm/yr). This precipitation gradient among the
3 locations would result in differences in pH which is known to be a key factor influencing bacterial
diversity [60,61]. In this study, we showed that location had a strong effect on bacterial diversity
and composition. We next address if there are any bacterial populations that were
disproportionately higher in relative abundance among the locations.

Our indicator taxon analyses identified 194 taxa that differed among the locations. The majority
of the taxa were associated with the wettest and driest locations (Carbondale ($n=181$); Hays
($n=151$)), whereas fewer were associated with the mesic intermediate location [Manhattan ($n=38$)]
(Supplementary Table S4), suggesting that Carbondale and Hays harbored higher number of
habitat specialist microbes due to the prevailing environmental conditions [62,63]. Of our three
common gardens, Manhattan had the most intermediate conditions for growth of *A. gerardii*,
where plants were less likely to experience abiotic stress including water availability stress in
Hays and water logged conditions in Carbondale. We further observed that the majority of soil
bacterial indicator ASVs were assigned to the following four phyla - Acidobacteria (Total $n=31$; C
$= 21$, H = 5, MHK = 5); Actinobacteria (Total $n=66$; C = 15, H = 47, MHK = 4); Planctomycetes

(Total n=20; C = 13, H = 7, MHK = 0); and Proteobacteria (Total n=109; C = 53, H = 41, MHK =
15); and Verrucomicrobia (Total n=23; C = 14, H = 9, MHK = 0) (Supplementary Table S4). It is
unsurprising that these phyla dominate in our indicator taxon analysis since they are ubiquitous
among rhizosphere associated bacterial communities [41,64-66]. However, these indicators were
not evenly distributed across locations, suggesting that this pattern was due to the soil moisture
affecting the bacterial distribution. Soil moisture can directly affect the soil microbial composition
and functionality by differentiating drought tolerance among taxonomic and functional groups of
microorganisms [67,68]. For example, limited soil moisture restricts the solute mobility and
therefore decreases substrate supply to the soil microbes. Therefore, highly moisture-sensitive
Gram-negative bacteria populations (Acidobacteria, Planctomycetes, Verrucomicrobia) were
affected by the limited water, resulting in lower number indicators in Hays and disproportionately
higher in Carbondale [69-71]. In line with this argument, we also observed that Gram-positive
bacteria (Actinobacteria) were disproportionately more abundant in Hays than in Carbondale and
Manhattan [72].

***Ecotypes shaped the rhizosphere inhibiting bacterial richness and diversity***

While there was a strong location effect on all alpha diversity metrics, a marginal ecotype effect
was observed only in (S_{Obs}) richness (ANOVA, S_{Obs} : $F_{2,245} = 3.46$; $P = 0.033$). The pairwise
comparisons among ecotypes indicated that the mesic ecotype rhizobiome had a higher observed
richness (S_{Obs}) than the wet ecotype (Mesic vs Wet: $H = 6.86$, $P_{adj} = 0.026$). We observed no
differences in bacterial diversity and phylogenetic diversity among host ecotypes (ANOVA,
Shannon's H' index: $F_{2,245} = 0.91$; $P = 0.404$; Faith's PD index: $F_{2,245} = 2.51$; $P = 0.108$) (Figure
2A). It was not surprising that the effect of ecotype was lower than the effect of location. It is well

[revised manuscript text omitted]

we noticed dissimilarities among locations and ecotypes across both *Pseudomonas*-Oligos and
*Rhizobium*-Oligos (Figure 4). Even though the relative abundance distribution patterns of oligos
was obviously driven by location, we observed differences in ecotypic recruitment across
ecotypes. Further work is necessary to provide more insights into recruitment patterns of strain-
level Pseudomonadales and Rhizobiales populations by the plant host to understand the impact
of bacterial populations on hosts' resilience to environmental stresses. However, our results
suggested that precipitation-induced stress (drier in Hays and wetter in Carbondale) could result
in a shift of plant host-rhizobiome interaction, changing the *Pseudomonas*-Oligos and *Rhizobium*-
Oligos composition. For example, in Hays, *Pseudomonas*-Oligos-GG were more prominent in
dry ecotype as compared to wet ecotype, while *Pseudomonas*-Oligos-AG were highly detected in
mesic ecotype (Figure 4A). On the other hand, in Manhattan, *Pseudomonas*-Oligos-GA
decreased in relative abundance in both dry and wet ecotypes in Manhattan, with an increase in
relative abundance of *Pseudomonas*-Oligos-AA. Similarly, in Carbondale, there was a higher
relative abundance of *Pseudomonas*-Oligos-AA wet ecotype as compared to dry ecotype (Figure
4A). We further noticed that in Hays, there was a higher relative abundance of *Rhizobium*-Oligos-
ATATC in dry ecotype as compared to wet ecotype. However, the relative abundance of
*Rhizobium*-Oligos-ATATC was higher in the wet ecotype as compared to dry ecotype in
Manhattan. Interesting *Rhizobium*-Oligos-TGATG was substantially higher in relative abundance
in wet and mesic ecotypes as compared to dry ecotyp in Carbondale (Figure 4B). The differences
in the *Pseudomonas*-Oligos and *Rhizobium*-Oligos across the locations and ecotypes potentially
support the "home field advantage" and tendency of plants to recruit microbial strains that could
contribute to the plant hosts' ecotypic resilience [21,84]. The intraspecific genome content may be
particularly important in understanding these host-microbe interactions.

**Conclusion**

Plant responses and adaptation to stress depend on a combination of environmental factors and
plant genetics [1,2]. While the environment defines the soil-associated microbial pool, our results
highlighted the plant-host's ability to recruit soil microbial communities under different
environmental conditions. We showed that plant hosts were more successful in the recruitment
of both the general and unique microbial population when grown at their homesite [85]. Our
observations also suggested the "*home field advantage*" and mutual association between plant
and specific soil microbes [21,22,84]. From our study, we further propose that, at their home sites,
ecotypes originating from these harsher environments, experiencing constant abiotic stresses,
were better able to recruit specialist microbes with the potential stress relief functions [85]. We
argue that in the study of plant host-rhizobiome, in line with the generalist and specialist
hypothesis, plant host resiliency under abiotic stress are more dependent on specialized microbes
[2,4,63,76,77] rather than the core microbiome [15,86-88], Our study suggests that in terms of the
resistance to abiotic stress, the key factor is in less abundant and specialized microbes with
specific functions rather than general common microbes which are abundant in soil. We further
observed, in our study, the fragile relationships between plant hosts and associated specialized
microbes. Although it might be challenging to tease apart the "*specialist vs generalists*" and "*home*
*field advantage*" concepts, we used oligotyping analysis and showed the ecotypic variations in
recruitment of bacterial stains from the same genera. These observations even further highlight
the complexity of these symbiotic relationships. After growing in a common garden for over 10
456 years, we could still see the distinct microbial communities recruited by the ecotypes. However,
the disconnect between the plant host ecotypes and rhizobiome in the harsher environment of
Hays demonstrated that the plant host-rhizobiome interaction is much more fragile than we
imagined. Taking all these factors in consideration, there is a crucial gap to identify
[revised manuscript text omitted]

- 39. Galliard M, Bello N, Knapp M, Poland J, St Amand P, Baer S, et al. Local adaptation,
genetic divergence, and experimental selection in a foundation grass across the US Great
Plains' climate gradient. *Glob Chang Biol*. 2019;25: 850–868.
- 40. Johnson LC, Galliard MB, Alsdurf JD, Maricle BR, Baer SG, Bello NM, et al. Reciprocal
transplant gardens as gold standard to detect local adaptation in grassland species: New
opportunities moving into the 21st century. *J Ecol*. 2021. doi:10.1111/1365-2745.13695
- 41. Sarkar S, Kamke A, Ward K, Rudick AK, Baer SG, Ran Q, et al. Bacterial but Not Fungal
Rhizosphere Community Composition Differ among Perennial Grass Ecotypes under
Abiotic Environmental Stress. *Microbiol Spectr*. 2022;10: e0239121.
- 42. Raffard A, Santoul F, Cucherousset J, Blanchet S. The community and ecosystem
consequences of intraspecific diversity: a meta-analysis. *Biol Rev Camb Philos Soc*.
2019;94: 648–661.
- 43. Kokko H, Chaturvedi A, Croll D, Fischer MC, Guillaume F, Karrenberg S, et al. Can
Evolution Supply What Ecology Demands? *Trends Ecol Evol*. 2017;32: 187–197.
- 44. Galliard M, Sabates S, Tetreault H, DeLaCruz A, Bryant J, Alsdurf J, et al. Adaptive genetic
potential and plasticity of trait variation in the foundation prairie grass *Andropogon gerardii*
across the US Great Plains' climate gradient: Implications for climate change and
restoration. *Evol Appl*. 2020;13: 2333–2356.
- 45. Smalla K, Wieland G, Buchner A, Zock A, Parzy J, Kaiser S, et al. Bulk and rhizosphere soil
bacterial communities studied by denaturing gradient gel electrophoresis: plant-dependent
enrichment and seasonal shifts revealed. *Appl Environ Microbiol*. 2001;67: 4742–4751.

- 46. Sanguin H, Remenant B, Dechesne A, Thioulouse J, Vogel TM, Nesme X, et al. Potential of
a 16S rRNA-based taxonomic microarray for analyzing the rhizosphere effects of maize on
*Agrobacterium* spp. and bacterial communities. *Appl Environ Microbiol.* 2006;72: 4302–
4312.
- 47. Bolyen E, Rideout JR, Dillon MR, Bokulich NA, Abnet CC, Al-Ghalith GA, et al.
Reproducible, interactive, scalable and extensible microbiome data science using QIIME 2.
*Nat Biotechnol.* 2019;37: 852–857.
- 48. Martin M. Cutadapt removes adapter sequences from high-throughput sequencing reads.
*EMBnet.journal.* 2011;17: 10–12.
- 49. Callahan BJ, McMurdie PJ, Rosen MJ, Han AW, Johnson AJA, Holmes SP. DADA2: High-
resolution sample inference from Illumina amplicon data. *Nat Methods.* 2016;13: 581–583.
- 50. Gihring TM, Green SJ, Schadt CW. Massively parallel rRNA gene sequencing exacerbates
the potential for biased community diversity comparisons due to variable library sizes.
*Environ Microbiol.* 2012;14: 285–290.
- 51. Paradis E, Schliep K. ape 5.0: an environment for modern phylogenetics and evolutionary
analyses in R. *Bioinformatics.* 2018;35: 526–528.
- 52. De Cáceres M, Legendre P. Associations between species and groups of sites: indices and
statistical inference. *Ecology.* 2009;90: 3566–3574.
- 53. Bates D, Mächler M, Bolker B, Walker S. Fitting Linear Mixed-Effects Models Using lme4. *J*
*Stat Softw.* 2015;67: 1–48.
- 54. McMurdie PJ, Holmes S. phyloseq: an R package for reproducible interactive analysis and
graphics of microbiome census data. *PLoS One.* 2013;8: e61217.
- 55. Oksanen J, Blanchet FG, Friendly M, Kindt R. vegan: Community Ecology Package. R
package version 2.5-6. 2019. 2020. Available:

[revised manuscript text omitted]

information is in Supplementary Table S5. Higher numbers of unique ASVs suggest the "*home*
*field advantage*" of the *Andropogon gerardii* ecotypes.

Figure 4. Relative abundance of each oligotype within the Pseudomonadales and Rhizobiales
diversity for each sample among three sites located across precipitation gradient. The proportion
of the relative abundance of the (A) Pseudomonadales (red, top) and *Pseudomonas*-Oligos
(bottom), and (B) Rhizobiales (purple) and *Rhizobium*-Oligos (bottom) suggest the impact of
locations and ecotypes on rhizobiome strain-level composition.

Table 1. Statistical analyses (ANOVA and pairwise test) reveal locations (C-Carbondale, H-Hays,
MHK-Manhattan) had a strong influence on the rhizobiome diversity and richness.

**Supplementary Tables**

Supplementary Table S1. Locations of seeds and populations collected from *Andropogon*
*gerardii*.

Supplementary Table S2. Samples names and details for the soil cores collected for this study.

Supplementary Table S3. Raw sequence analysis by QIIME 2 Version 2019.7. The number of
counts of bacteria and fungi initially obtained, and the counts that were considered after primer
trimming and DADA2 quality control per sample. Bacterial identifications along with the number
of counts per sample.

Supplementary Table S4. Indicator taxon analysis on different locations (irrespective of ecotypes)
- Hays, Manhattan and Carbondale.

Supplementary Table S5. Unique taxa for locations and ecotypes.

Supplementary Table S6. Indicator taxon analysis on ecotypes (irrespective of locations) - Dry,
Mesic and Wet.

Supplementary Table S7. Indicator taxon analysis on locations by individual ecotypes.

Supplementary Table S8. Indicator taxon analysis on ecotypes by individual locations.

Supplementary Table S9. Genus-level oligotyping analysis.

Figure 1

Figure 2

Figure 3

Figure 4

Table 1

Index	ANOVA		Pairwise test		
	chi-squared	P-value	C	H	
SObs	10.42	0.005	0.1005	-	H
			0.0067	0.0653	MHK
Shannon's H	25.171	3.422e-06	0.7	-	H
			6.90E-05	1.40E-05	MHK
Faith's PD	6.7829	0.03366	0.044	-	H
			0.044	0.585	MHK

Index	ANOVA		Pairwise test		
	chi-squared	P-value	Dry	Mesic	
SObs	3.46	0.033	0.585	-	Mesic
			0.078	0.026	Wet
Shannon's H	0.91	0.404	0.285	-	Mesic
			0.285	0.061	Wet
Faith's PD	2.51	0.0108	0.78	-	Mesic
			0.14	0.13	Wet

Dear Dr. Renee Arias
Editor, Microbiology Spectrum

We are thankful to the reviewers for their suggestions and comments. We have made extensive changes to the manuscript (highlighted in blue) based on their comments. Briefly, we

- Adjusted our language to ensure clarity of our data analyses as well as ensuring our discussions and conclusions reflect our data.
- We also added the soil biochemistry total carbon and total nitrogen analyses, as well as pH, soil moisture and texture to provide more insights into our study.

We have included a revised “clean” and “marked up” version for your perusal. Our responses to the reviewers’ comments are outlined in “blue”. The line numbers mentioned here correspond to the source text file of the “clean” version. We hope that you and the reviewers will agree that changes to our manuscript provide more clarity of our work to the readers of Microbiology Spectrum.

Reviewer #1 (Comments for the Author):

1. L141. How were the populations different genotypically? It appears from the reference, that the seeds were obtained on site at each location and referred to as populations from each of the three locations. How different were these populations from each other and especially among ecotypes? This would be important background information. Currently we have no idea about the genotypic differences associated with the treatment experimental design.

The plant populations vary among but do not differ within the ecotypes. This was shown in Johnson et al. 2015[1]. We have added below text to provide clarity to the information on the ecotypic populations:

Ln 114 - 123: Briefly, in 2009, seeds were collected from four populations, which jointly defined each of the three regional ecotypes. Morphological traits of the populations vary among ecotypes, but populations do not differ within ecotypes, thus confirming the regional nature of the ecotypes [45–48]. Seeds were collected from mixed grass prairie in Central Kansas (referred as CKS/Dry: CDB, REL, SAL, WEB), and tallgrass prairie in Eastern Kansas (EKS/Mesic: CAR, KON, TAL, TOW) and Southern Illinois savanna (SIL/Wet: 12MI, DES, FUL, WAL) (Acronyms of populations are listed in Supplementary Table S1) across the natural rainfall gradient with 580 mm/year, 871 mm/year and 1167 mm/year of mean annual precipitation, respectively. Selected

populations originated from intact prairies within a 80 km radius of each reciprocal garden site [47].

2. L148 The seeds from each population were first germinated in potting soil mix in a greenhouse environment. Then the seedlings were transported to the field. Why were they not established directly in the field? Could the establishment method have introduced an artificial aspect into the rhizobiome prior to sampling, given their perennial nature? How long were the plants established before sampling took place? This is not indicated at all. If everything occurred in the first year then there may be a problem.

We thank the reviewer for raising an important concern on the potential introduction of artificial effects on the plant microbiome from the greenhouse germination. We used the greenhouse approach to ensure the establishment of the plants on the field at the same time. The germination rate of *A. gerardii* is 60% which might lead to difficulties in the single plant establishment on the field. Therefore, we germinated seeds in the potting mix and transplanted 3- to 4- month-old seedlings to the field. Additionally, the reciprocal gardens were established in 2009 (L134 in text) and the sampling took place in 2019 (L133), almost 10 years after the establishment. We believe that after 10 years in the common garden, the artificial aspects potentially introduced by the greenhouse effect, would become insignificant. We have added the following to the revised manuscript for clarity.

Ln 123 - 125: Seeds representing the three ecotypes and twelve populations were germinated and grown in a greenhouse using potting mix (Metro-Mix 510)

Ln 129 - 130: After almost 10 years after establishment in 2009, we sampled *A. gerardii* reciprocal gardens for this experiment.

3. Also the work represents only a single year of data. In almost all field studies multiple years are required to come to a certain conclusion.

We agree with the reviewer that observing a multi-year data set would complement our conclusion. Like many perennial grasses, *A. gerardii*'s lifespan is ~30 years with the age of maturity is ~5 years [2]. After 10 years since establishment, mature plants would not undergo the developmental modification that might significantly affect microbial communities from year to year. Therefore, we believe that one year of samples was sufficient in order to answer **our research question whether there is an ecotypic effect on the rhizobiome**. We are in the process of collecting and archiving multiple years rhizobiome to provide insights into how ecotypes may shape their rhizobiome in a time series. Although that is not within the scope of this manuscript, we are excited to provide insights into plant host-rhizobiome interaction in the future.

4. At what time of year were the plants sampled. What were the conditions at and prior to sampling in terms of soil moisture and other factors for each site? Especially for soil moisture. What are the characteristics of the soils at each location is left completely unstated in terms of soil type, soil structure, and soil nutrients. These may present alternative explanations associated with the findings.

We thank the reviewer for their comment. Absolutely, the environmental factors such as the soil moisture and the time of the sampling can greatly affect the microbial communities. To address your comment, we performed and analyzed additional soil chemistry analysis on the soil pH, %C and %N. In addition, we identified the soil texture and soil moisture content at each location. The data results are presented in Figure 1, Supplementary Table S4, and discussed in the new section “Differences in precipitation resulted in the 
[revised manuscript text omitted]

5. How was the sampling performed besides just coring the soil? What location in relationship to the plant root system? Was a single core obtained from each plant or multiple cores. Were these cores bulked and then subsampled? Single cores would increase experimental variability, bulking would produce a more representative sample. Also, the 15 cm cores are likely only to contain soil from surface roots, but given the deep rooting nature of big bluestem does such a core really represent the plants rhizobiome?

We agree with the reviewer that sampling details on the are necessary for understanding the study outcomes. Additional information on the sampling approach was added to the text for transparency and clarity. Single (15 cm x 1.25 cm diameter) core was sampled from each plant. We collected soil cores by placing the core as close to the plant as possible. The side of the plant was picked randomly. Each core was treated as

independent treatment and was never combined with other cores. The top layer of soil organic matter (top 15 cm) is a standard for microbial dynamics studies due to the highest microbial activity in this layer of soil. Soil cores were collected in the ziplock bags, placed in the cooler with ice and transported to the laboratory. Roots were picked manually, shaken from the excess soil and weighed for the DNA extraction. We have added the following in the revised manuscript:

Ln 132 -136: To track microbial communities, we collected the top-most soil layer (top 15 cm) due to the highest microbial activity in this layer of soil. We collected a single core (15 cm deep x 1.25 cm diameter) as close as possible to each plant. The side of the plant was picked randomly to ensure sample heterogeneity. Each core represented an independent sample, i.e., cores were not pooled with other cores.

Ln 145 - 149: We extracted total DNA from 0.150 g of roots and rhizosphere soil from all 284 samples using an Omega E.Z.N.A. Soil DNA Kit (Omega Bio-Tek, Inc., Norcross, GA, USA) as per the manufacturer's protocol with a slight modification. Roots were picked manually from the collected soil cores, shaken from the excess soil and weighed for the DNA extractions. Any soil that remained attached to the roots was considered rhizosphere soil [49,50].

6. L155 Some plants died during establishment. How did this affect the sampling distribution among sample cores? Did any ecotype population suffer from this dieback? This needs to be stated

We thank the reviewer for raising this great point. Thank you for bringing this up, because the death of samples from one group over another might lead to issues in interpreting the data. We re-assessed our dataset and observed that there were only seven plants that died and the rest of the samples had failed the amplification. We added additional details and clarifications in the text of the revised manuscript:

Ln 136 - 140: Seven plants had not survived the transplanting or through the ten years in the common gardens. However, the plant mortality differed in the Pearson's chi-squared test neither for ecotype (χ^2 -squared = 4.44, df = 2, p-value = 0.109) nor 138 population (χ^2 -squared = 9.43, df = 12, p-value = 0.665), suggesting that mortality was random without predictable patterns across ecotypes or populations.

7. L160 Did the authors extract DNA and sequence all 284 cores? How does the sampling strategy feed into the DNA sequencing?

Yes, all the collected soil cores were treated as independent treatments and were extracted, sequenced and analyzed separately. We added the following text in the revised manuscript for clarity:

Ln 145 - 147: We extracted total DNA from 0.150 g of rhizospheric roots and soil from all 284 samples using an Omega E.Z.N.A. Soil DNA Kit (Omega Bio-Tek, Inc., Norcross, GA, USA) as per the manufacturer's protocol with a slight modification.

8. L164 Since the authors extracted the roots from the soil cores and shook off the non rhizosphere soils and then extracted the DNA then the results represent not only the rhizosphere but also the endorhizosphere organisms. This should be stated.

We believe that this is a valid concern of the reviewer about including the endorhizosphere microorganisms. That is true, that the DNA extraction kit Omega E.Z.N.A. Soil DNA Kit (Omega Bio-Tek, Inc., Norcross, GA, USA) we are using included a bead beating step. However, the bead beating was sufficient for lysing bacterial cells, but was not enough (duration and strength) for the plant root cells. Therefore, we were extracting rhizosphere and rhizosphere soil microorganisms. In order to extract endorhizosphere organisms, the root grinding with liquid nitrogen would be required [3]. We added the following in the revised manuscript:

Ln 151 - 154: Even though the mechanical lysis was a step in the DNA extraction process, the grass roots were never not fully grinded in the process, therefore our results represent only rhizosphere microorganisms, but not the endorhizosphere organisms.

9. L200 The authors performed indicator analysis using a wide range of computer algorithms, but it is unclear exactly how this was done. More detail would be appreciated. Following with the ANOVA models, we used the ASVs data to check if there were bacterial populations that were more abundant in one group than in another. We added the following information in the revised manuscript for clarity:

Ln 231 - 234: To determine if any sites, ecotypes, or ecotypes within each of the sites had Amplicon Sequence Variants (ASVs) that were disproportionately more abundant in one group than in another, we used indicator species analysis with the "multipatt()" function in R package "indispecies" (v. 1.7.12) [60].

10. L217 The authors performed analysis on 110 samples. (2,104,416/19,135) How does this relate to the 284 plants? Again there is confusion concerning the sampling system? See L160 comment

We are really appreciative of the thorough review of the text by the reviewer. We have made a mistake in reporting this data. The reported number only reflects the number of sequences for Hays location (Supplementary Table S3_ASVs assignments). The supplementary data was reporting the correct information, but we have made a mistake in

the text. We have corrected the information in the revised manuscript to reflect on the correct data:

Ln 249 - 250: We analyzed a total of 5,628,302 ($19,818 \pm 4531$ per sample) high quality sequences assigned to 1,441 ASVs. (phyloseq v 1.42.0 [63], R ape [64]).

11. L233 The table only presents p values and not absolute values for each of the indices so we really can't judge whether one indices is higher or not.

Thank you for your comment, even though the absolute numbers are present in the text of the manuscript, it is beneficial to add this information in the visualization table. We have added the additional information in the revised Table 2.

12. L234 The data only supports a lower diversity part of the definition.

13. L236-237, 248 intermediate conditions in terms of what factor? There are likely multiple factors working on this system and the authors only focus on moisture. That is not very convincing.

The reviewer is correct that the statement about the good colonization and fast growing microbes is speculation. We do not have quantified data to support this statement. We have adjusted our statement in the revised manuscript to more accurately reflect our data:

The reviewer is correct that soil moisture is not the only factor affecting the microbial community diversity and composition. In order to give more insights on the environmental conditions across location we added the soil chemistry, soil texture and water content analysis to our manuscript. According to that data the environment of Manhattan is less extreme when compared to Hays and Carbondale for plant growth. We have added the following sentence to the revised manuscript to enhance clarity for the readers:

Ln 322 - 324: Phylogenetic diversity analyses suggested that Manhattan was favorable for generalist microbes characterized by the lower diversity [2,4] due to the intermediate soil chemistry and moisture conditions as compared to Hays and Carbondale.

14. L251 pH is certainly a factor that affects bacterial diversity in soils. The authors do not present any evidence that pH differed in the soils. The idea presented here is that drier soil due to lower moisture increases pH. There are other factors that are more important than moisture that determines soil pH such as soil substrate. Soil data should be presented
Thank you for your comment. Absolutely, there are many more factors like climate, mineral content or soil texture that can affect soil pH. As described above, we performed and analyzed additional soil chemistry analysis on the soil pH, %C and %N. In addition, we identify the soil texture and soil moisture content at each location. The data results are

presented in the Figure 1, Supplementary Table S4, as well as discussed in the section “Differences in precipitation resulted in the distinct soil chemistry and texture parameters across sites” section of the updated manuscript.

15. L 263 were the soil waterlogged at time of sampling. No data presented.

The soil was not waterlogged at any of the locations during our time of sampling. However, we have performed soil moisture content analyses in order to quantify our statement. The additional information is added to the text, figures and supplementary material in the revised manuscript. See Figure 1, Supplementary Table S4

16. L308 The numbers in the Venn diagram are used to indicate the home field advantage. It would be much better if there were statistics attached to these numbers to determine if there was a statistical difference. As is, there is only one replication which does not lend support for the contention for the most important objective of this work.

Venn diagrams were used to show the unique and shared rhizosphere community members across locations, ecotypes, and ecotypes at each location. Venn diagrams do not represent the richness of the microbial populations, but look at the presence vs absence for each factor. Each microbial member was counted as 1 for the presence and 0 for the absence. The overlap between all groups would count as three and represent the shared middle of the Venn diagram. The unique microbes associated with only one group would be a non-shared part of the Venn diagram. Our Venn diagram representation of the data showed that ecotypes recruit more unique microbes when grown at “home” site suggesting the “home-field advantage”. The data visualization method does not require statistical analysis, but was used to show the pattern across our data. Additional information/ values for the Venn diagrams are in Supplementary Table S6 for the readers.

17. L328 The authors state that increased photosynthetic rates under dry conditions would favor microbial specialists. Again this is sheer speculation without any supporting data, only a reference. The sentence should be phrased as such.

The corrections were made in the revised manuscript:

Ln 416 - 418: On the other hand, the dry ecotype in Hays produced lesser biomass but maintained higher photosynthetic rates, which potentially could result in continuous and steady supply of photosynthate in limited quantities favoring slow growing microbial specialists [76].

18. L358 Why are there no indicator species in the dry habitat?

Similar to the community analysis of our data, the indicator taxon analysis among ecotypes was dominated by the location effect. In addition, in our indicator taxon analysis we used $\alpha = 0.005$ to ensure robustness in identifying the indicator species. Although

this alpha shows the microbial indicators across groups, it is less sensitive to the comparatively rare or a lower in number microbial taxa across. To focus on the rare taxa, which we suspect are specialist microbes, we manually calculated the unique microbes across locations and ecotypes. We have added the following text in the revised manuscript for clarity:

Ln 446: The alpha = 0.005 was added in the text

19. L392 This data since it is discussed should be presented as a figure or table rather than relegated to the supplement.

Similar to the previous comment, we believe the indicator taxon analysis was not sensitive enough to identify rare specialists. Although we briefly discussed about “rare specialists” to illustrate the importance of paying attention to these group of microbes in understanding plant host-rhizobiome interaction, we want to ensure focus of the manuscript on our research question answering if ecotypes have an impact on their rhizobiome. Therefore, we believe that this indicator taxon analysis would not bring new information to the manuscript but dilute the vision of the story to focus. However, we want to be transparent in our reporting, and thus have this information in the Supplementary (Supplementary Table S7, S8 and S9) for interested readers.

20. Table 1 is out of focus. Should be a sharper image. Also it presents only p values. The table could be reconstructed to present both absolute values as well as P values. We have redo Table 2 to include the absolute values and P values, as well as output a higher resolution image.

Reviewer #2 (Comments for the Author):

1. Ln41. I am not sure I understand the sentence, if possible, rewrite '..data also suggest that ecotypes were more successful in recruiting rhizosphere community members unique to their local homesites'. Does this mean that when the ecotype, whose seed was collected on a mesic site, was planted at the mesic site, then a greater number of bacterial taxa unique to the ecotype and site combination were recovered?

We thank the reviewer for their suggestion. We rewrote the sentence to make it clearer. In our results we see the evidence that ecotypes are able to recruit the microorganisms that are unique to their homesites. For example, the dry ecotype was more successful at the recruitment of unique to the dry location microbes when compared to wet and mesic ecotypes planted at the same location. We have edited the following sentence in the revised manuscript to improve clarity:

Ln 10 - 13: Our data also suggest that ecotypes planted at their homesites were more successful in recruiting rhizosphere community members that were unique to the location. The link between the plants' homesite and the specific local microbes supported the "home field advantage" hypothesis.

2. Ln 44. The evidence supports that unique microbes could be related to adaptation to sites with variations in water availability. No data is presented regarding plant stress responses. I recommend removing the statement regarding plant stress responses.

We agree with the reviewer that the water availability and other environmental characteristics such as soil texture or soil pH would influence the microbial composition across sites. Our work found evidence that ecotypes at home locations, especially in harsh environments, have a pattern of selection specific to home microbes suggesting the "home field advantage". We speculate that by selecting "specialized" microbes that are able to survive in extreme conditions, local ecotype gets an advantage. In addition, we also provided additional quantitative evidence in regard to water availability in the revised manuscript. We added numerous additional soil chemistry and water availability (%C, %N, pH, soil moisture, soil texture). These data are reflected in the revised manuscript in Figure 1, Supplementary Table S4 and in the additional section: "Differences in precipitation resulted in the distinct soil chemistry and texture parameters across sites" of the revised manuscript.

3. Ln 48-50. The last two sentences of the abstract could be rewritten for clarity. For example, Ln 61-62 is a similar statement and easier to understand.

We thank the reviewer for the suggestion. We have revised the last two sentences of the abstract in the revised manuscript:

Ln 17 - 21: These results improve our understanding of the complex plant host-soil microbes interactions and should facilitate further studies focused on exploring the functional potential of recruited microbes. Our study has the potential to aid in predicting grassland ecosystem responses to climate change and impact restoration management practices to promote its sustainability.

Introduction

4. Ln 92. Given the emphasis on "home field advantage" in this manuscript, I recommend including definitions and examples of these.

We added the definition to the revised manuscript:

Ln 63 - 65: In previous studies, the "home field advantage" is described as stability of the performance of an organism at the "home" environment, and decrease in performance away from "home" [21–23].

5. Ln 119-122. Are (1) and (2) in this paragraph different objectives? It seems that (1) states the objective and (2) the approach. In this work, the authors investigated the "relative importance of the plant local environment and phenotypic variation ...on establishing plant rhizobiome" by comparing rhizosphere bacterial communities "under reciprocally transplanted, local and non-local adapted environments".

Objectives 1 and 2 try to answer different questions. We made changes to the revised manuscript to improve clarity:

Ln 94 - 96: Here, we (1) investigated the relative importance of the plant environment and phenotypic variation of *A. gerardii* on establishing plant rhizobiome; (2) compared three regional *A. gerardii* ecotypes planted reciprocally in their ability to recruit microbes in local and not-local environments

6. Ln 122. Number (3) indicates examining potential links between members of the bacterial community and plant hosts' resilience characteristics. There is no evidence presented in this article that relates to specific plant traits that could contribute to adaptation to different soil moisture environments, nor analysis performed to determine the relationship between specific plant traits and bacterial communities. In different places in the manuscript, prior studies are referenced, which do summarize specific traits identified in individual ecotypes. But the current analysis focuses on ecotypes and the location of common gardens.

We thank the reviewer for the comment. We agree, and have removed objective 3 to clearly enhance the focus of the manuscript.

7. Ln 178-179. By runs, do you mean sequencing runs? If so, how many sequencing runs were needed to process the analyzed samples? Also, what were the parameters used for the DADA2 pipeline. Provide a similar level of detail as it was provided for the section on primer trimming.

We added details to the revised manuscript to provide more information:

Ln 199 - 213: Following initial quality control, we used DADA2 [55] with the same parameters across 2 different runs and truncated the reads to length where the 25th percentile of reads had a quality score below 15 (Forward 231 and Reverse 229). The first run included 24 samples and was used as a trial run for the project, and the rest of the project samples were sequenced when the quality of the samples was confirmed. Since the same primers were used for the first 24 samples, the quality control and primer removal using DADA2 were performed separately. Samples then were merged together (total n = 284) and analyzed as one dataset. We used the pre-trained SILVA database (v. 138) in QIIME 2 for taxonomic assignment of the bacteria. Sequences were blinded to amplicon sequence variants (ASVs) and any unknown or unclassified ASVs were removed from downstream analysis. We rarefied the data set to 10,000 reads per sample (resulting in 2,104,416 high quality sequences) to minimize biases resulting from differences in sequencing depth among samples before estimating diversity indices and downstream analyses [56]. We used QIIME 2 for the sequence processing pipeline, whereas the subsequent statistical analyses for microbial richness, diversity and composition were performed using R studio [57].

8. Ln. 179. What is the justification for rarefying the data set to 10,000 sequences per sample (and not 20,000 or 50,000 sequences per sample, for example)? By briefly looking at the supplementary tables, it seems a higher number could have been used and still not lose too many samples with low reads. Also, is it appropriate to assume that rare taxa (which will be lost by rarifying to 10,000) have less relevance for studies on local adaptation or home advantage? Will higher sequencing depth per sample result in similar results?

We thank you and agree with the reviewer that higher sequencing depth will reveal rare taxa. However, we took multiple considerations on determining the rarefaction at 10,000 reads/sample. We checked the alpha diversity, including observed_features, faith_pd, and shannon diversity at different rarefaction depths. As the attached figure (top figure) shown in alpha-rarefaction.qzv file shown (uploaded in figshare 10.6084/m9.figshare.21791306), the change of the indexes is not significant after depth reaches ~10,000 reads/sample. This result indicates that higher sequencing depth will not significantly affect the alpha diversity. More importantly, if we choose 20,000 reads/sample, we will lose 33%-55% of samples per location in our analysis (figure bottom), which will affect the power of statistical analysis in our study. As such, we are

confident that the choice of 10,000 reads/sample is justified for this study in elucidating the interactive relationship between plant host and its rhizobiome.

9. Ln 184-202. Indicate which portions of the analysis were performed in Qiime2, and which in R. For specific analysis include the functions used, and when appropriate indicate references. For example, is the betadispr function a function from which package, and is there a reference that can be used to replicate these analyses?

We made edits to the statistical approach in the revised manuscript to ensure clarity for the readers:

1) Ln 211 - 213: We used QIIME 2 for the sequence processing pipeline, whereas the subsequent statistical analyses for microbial richness, diversity and composition were performed using R studio [57].

2) Ln 219 - 221: Following overall ANOVA, we used pairwise comparisons using Kruskal-Wallis test to identify factors that were driving the significant effects (R studio v 4.1.1)[52].

3) Ln 223 - 234: We used the vegan package [58] to estimate the pairwise Bray-Curtis distances to compare the bacterial communities among the different factors. We then used the ggplot2 package [59] to visualize these data using Non-Metric Multidimensional Scaling (NMDS) ordinations. We used a nonparametric PERMANOVA to determine whether bacterial communities differed compositionally among sites, ecotypes and populations as well as their two- and three-way interactions. In this model, “population” was nested within “ecotype.” Following the PERMANOVA, we performed pairwise comparisons using the pairwise.adonis function. We also used betadispr function in the vegan package [58] to test whether the community dispersion differed between any significant groups. To determine if any sites, ecotypes, or ecotypes within each of the sites had Amplicon Sequence Variants (ASVs) that were disproportionately more abundant in one group than in another, we used indicator species analysis with the “multipatt()” function in R package “indispecies” (v. 1.7.12) [60] .

10. Ln 213. What were the relative abundances of Pseudomonas and Rhizobium? Also, why only high abundance taxa were selected? Could it be possible that less abundant or rare taxa be relevant for this study? Finally, in this section the genus name is used, however, the discussion and figures are at the order level (Pseudomonadales and Rhizobiales), not the genus level (Pseudomonas and Rhizobium). For consistency, I recommend using the order in this section as well.

The relative abundances of Pseudomonadales and Rhizobiales were $12.9 \pm 19.0\%$ and $11.9 \pm 3.6\%$ respectively. Pseudomonadales and Rhizobiales were chosen because not only they had the highest relative abundance, but also displayed interesting opposite relative abundance across the sites. The goal of displaying these two oligotypes is to 1) potentially support the “home field advantage” and tendency of plants to recruit microbial strains that could contribute to the plant hosts’ ecotypic resilience; 2) highlight the intraspecific genome content may be particularly important in understanding these host-microbe interactions. However, to be transparent and provide readers with the options, we have updated Supplementary Table S10 with all the oligotypes identified in this study.

We admit that less abundant or rare taxa could be relevant for this study, but we are not 100% confident with the rare taxa contributing to this study given the amount of coverage in the sequencing reads. Although not part of this manuscript, we are equally excited about the reviewer’s comment, and are already in the middle of a larger experiment using culturing and full genomic sequencing methods to elucidate the contribution of rare taxa to the plant host-microbe interaction. Stay tuned for that in the future!

We have made changes to use order nomenclature (Pseudomonadales and Rhizobiales) in this section for consistency.

Results and discussion.

11. Be consistent in the use of the terms site or location, but not the combination of 'site locations'. Also, be consistent in the use of these terms in the figures and legends. For example, Figure 1 legend uses the word site in Ln 723, then location in 725, 'site location' in 727, and location in the figure key. In general, this section is difficult to read, with results of different analyses (ANOVA, PERMANOVA, unique taxa, and indicator taxa) presented in different sections, and not a clear synthesis.

We have made changes to use the term “site” throughout the manuscript, figures and legends for consistency.

12. Ln 219 - Add a reference for the classifier tool used

The citation was added to the text:

Ln 250 - 251: Of the recovered reads, an average of 90% was annotated to the genus level using the Naive Bayesian classifier [65].

13. Ln 220 - Were plant reads (chloroplast or mitochondria) also removed from analyses?

Yes, only bacterial ASVs were included in the analysis. We have added the following sentence to the manuscript for clarity:

Ln 251 - 252: Any reads assigned to chloroplasts and mitochondria as well as any unknown or unclassified ASVs were removed from downstream analyses.

14. Ln 227 - What were the results of the full test, including both main effects and interactions? The materials and methods (Ln 184) state that the analysis was used to test the main and interactive effects in the various diversity indexes tested. Were there significant effects for the interactions between location, ecotype, and population?

Yes, all overall models included two way and three way interactions of factors. The model was $Y = \text{location} + \text{ecotype} + \text{population within ecotype} + \text{location*ecotype} + \text{ecotype*population within ecotype} + \text{location*ecotype*population within ecotype}$. The population combinations and all the interaction effects were not significant. In the manuscript we did follow-up statistics, and focused on the location and ecotype effect because they were significant in the overall ANOVA/PERMANOVA tests.

15. Ln 234 - How did you determine that a location was more favorable for generalist, fast-growing microbes? It is not clear how phylogenetic diversity data can be used to infer this.

The added soil chemistry, texture as well as %N and %C show that Manhattan had a most intermediate environmental condition when compared to Hays and Carbondale.

Therefore, we believe, these conditions would promote the growth of generalist microbes and dominate the Manhattan soil. However, we are also aware of the limitations of the data, and made changes to the sentence to reflect more accurately our results:

Ln 322 - 324: Phylogenetic diversity analyses suggested that Manhattan was favorable for generalist microbes characterized by the lower diversity [2,4] due to the intermediate soil chemistry and moisture conditions as compared to Hays and Carbondale.

16. Ln 240-246. Did the PERMANOVA model include location*Ecotype interaction?

Yes, all models included two way and three way interactions of factors. The model was $Y = \text{location} + \text{ecotype} + \text{population within ecotype} + \text{location*ecotype} + \text{ecotype*population within ecotype} + \text{location*ecotype*population within ecotype}$. We edited the following sentence to enhance clarity for the readers:

Ln 225 - 227: We used a nonparametric PERMANOVA to determine whether bacterial communities differed compositionally among sites, ecotypes and populations as well as their two- and three-way interactions.

17. Ln 250-254. In addition to precipitation gradient and pH are there other environmental parameters that vary across sites that could influence the results?

We appreciate the reviewer's comment and absolutely, there are many more factors like climate, mineral content or soil texture that can affect soil pH. To address their comment we performed the soil chemistry analysis on the soil pH, %C and %N. In addition, we identified the soil texture and soil moisture content at each location. The data results are presented in the Figure 1 as well as "Differences in precipitation resulted in the distinct soil chemistry and texture parameters across sites" section of the updated manuscript:

This precipitation gradient as well as difference in the soil chemistry and texture among the 3 sites would result in differences in pH which is known to be a key factor influencing bacterial diversity [4,5](Figure 1, Supplementary Table S4).

18. Ln 294. Why is a different ANOVA run? Were the ecotype and populations not included in the initial analysis?

As stated in the Ln 281 - 284: we observed a very strong effect of the location. This model included the factors: location, ecotype, population within ecotype and their interactions. In order to focus more deeply on the ecotype, for the second test, we split the data by the location and look at the ecotype and population effect at each location separately. For the second model we used factors: ecotype, population within ecotype and their interactions. We ran this model three times for Carbondale, Hays and Manhattan separately. We edited the manuscript to provide a clearer explanation for the readers:

Ln 381 - 384: To do this, we split our data set by sites, and performed separate ANOVAs for each site with ecotype, population and their interaction as the explanatory factors. We used this model separately for Carbondale, Hays and Manhattan, to focus on the effect of ecotypes at home and away from home sites.

19. Ln 308. Do unique taxa refer to taxa that are only present on an ecotype regardless of the population?

Since the effect of population was never significant in any tests across all the locations and each location separately in both the alpha diversity and community composition data, we did not include the population effect in the further deeper analysis of the unique taxa.

20. Ln 307-322. It is not clear how the bacterial taxa are considered either generalist or specialist, or "good plant colonizers"

We are very excited that the reviewer is asking the same types of questions we are trying to find answers to. In this study, we defined generalist microbes as fast growing microbes that are abundant in rhizosphere soil [6–8]. While generalist microbes are abundant in the soil with the mild condition of Manhattan, specialist microbes can adapt to the extreme environments of Hays and Carbondale. Our research showed the clear pattern of microbial distribution across location and ecotypes suggesting the “home field advantage”. However, in the scope of our current project with the resolution of 16S amplicon, we are not able to point out a certain taxa that will be defined as generalists or specialists. We are looking forward to sharing more data on this exciting subject in the near future. The next project that we are working on right now is specifically focused on teasing apart the generalist and specialist microbes and the differences in their functional potential using the metagenomic approach.

21. Ln 320-321. Did the referenced studies (63 and 36) document adaptation to environments with different water availability, or studies that focus on tolerance to drought stress? Do these report quantitative differences in traits in the different environments?

The paper (36) [1] documents the plant adaptation to the environment. The project was done on the same plots as used for the current project. This project describes the *A. gerardii* response under different climatic conditions, especially precipitation. According to the project findings due to the local adaptation ecotypes differ in vegetative cover, chlorophyll absorbance as well as gas exchange. This project report quantitative differences in traits in the different environments

The paper (63) [8] described the impacts of the environmental factors across seven climate types on the generalist and the specialist distribution and their effect on the community composition and dynamics. The paper describes the broad distribution of the generalists and their higher tolerance to environmental changes.

We made changes to the following sentences to provide more information for the readers:

Ln 410 - 415: Due to the physiological local adaptation to the wetter environment, wet ecotype might be poorly adapted to the drier environment of Hays - resulting in the lower photosynthetic rate [37]. We surmise that lesser volume of photosynthetic products exuded in the soil by the wet ecotype in Hays would limit the microbial taxa the wet ecotype could support in the drier environment [91]. In addition, the limited photosynthates exuded by the wet ecotype would then be metabolized by the more abundant [76] and faster colonizing generalists [92,93].

22. Ln 321. Include references that support the statement that lower photosynthetic rate results in lower amounts of root exudates or other deposits that could influence microbial recruitment in the root.

We added this citation to the revised manuscript. Ma et al. [9] reviewed effects of root exudates on the plant-microbe interactions. About 20% of all photosynthetically fixed carbon is released in the soil in the form of root exudates. Root exudates contain the soluble organic compounds that are used as a carbon source by the microbial communities. In addition to the carbon as a food source, root exudates contain non-volatile compounds that act as communication signals between plants and microbes. Therefore, the plants with lower photosynthetic rate will have a lower amount of carbon deposits to the soil which can influence the microbial recruitment by the plant.

Ln 412 - 414: We surmise that lesser volume of photosynthetic products exuded in the soil by the wet ecotype in Hays would limit the microbial taxa the wet ecotype could support in the drier environment [91].

23. Ln 349-354. Should this paragraph be in the previous section? Both the analysis of unique taxa and indicator species contribute to identifying taxa that could be associated with either (or both) location and ecotype.

We thank the reviewer for the suggestion. We further agree with the reviewer that both unique taxa and indicator taxa analysis arrive at the conclusion of “home field advantage”. However, we feel that the current way the manuscript provides a logical flow for the readers. This is especially true with the addition of the soil chemistry data. The flow of the manuscript was structured for the following reason - 1) We first provided the overall view of the unique taxa to make a point of not adequate sensitivity of the richness and diversity analysis. 2) After making this point with the unique taxa, we then move to the deeper statistical analysis of indicator taxa and oligos analysis to provide more evidence to the reader. 3) Although both unique taxa and indicator taxon analysis were performed for location, ecotypes and ecotypes within each location, the main point we try to express in this manuscript is the evidence of the mutual recognition between the ecotype and local microbes at the ecotype home location. Therefore, we believe in order to convey this message clearly to the readers, it is important to maintain the logical flow of the story, thus, the unique taxa analysis should stay as a part of the previous section and maintain a link between ideas. We hope the reviewer will see our logic in this argument.

24. Ln 356-366. Are the PERMANOVA results here different from the PERMANOVA results discussed in the prior section (Ln 303)?

Similar point #18 that was raised by the reviewer, this PERMANOVA test was done on each location separately to highlight the effect of ecotypes. The model was: $Y = \text{ecotype} + \text{population within ecotype} + \text{their interactions}$. We ran this model three times for Carbondale, Hays and Manhattan separately. We thank you the reviewer in raising this question, and have added some details to the revised manuscript to enhance readability for the readers:

Ln 450-455: Similarly, taking into consideration the strong effects of sites, we further focused our analysis by studying the ecotypic effects at different sites. We split our data set by sites, and performed separate PERMANOVAs for each site with ecotype, population and their interaction as the explanatory factors. We used this model separately for Carbondale, Hays and Manhattan, to focus on the effect of ecotypes at home and away from home sites.

25. Ln 406-409. The section begins with Pseudomonas and Rhizobium, and continues the discussion at the order level. Be consistent in the taxonomic level used for analysis and discussion.

We have made changes to use order nomenclature (Pseudomonadales and Rhizobiales) in this section for consistency.

26. Ln 404-434. This section focuses on the analysis of specific oligos or sequence types belonging to the Pseudomonadales and Rhizobiales. Are there statistical tests that can be applied to these results to provide support to the statements based on the visualization of sequence type distribution across locations and ecotypes (and their interaction)? Although this section was focussed on the analyses of Pseudomonadales and Rhizobiales, we did not perform any statistical analyses other than description of distribution patterns across locations and ecotypes. The rationale behind the oligotypic analyses was to provide the readers additional insights into the possible reasons and a call for action in needing a higher resolution data to clearly understand the mechanism behind the plant host-microbe interaction. We did not make a quantitative conclusion nor statistical analyses based on oligotypes, however, these two oligotypes satisfied our two criteria before we carried out the oligotype analyses - 1) high ASVs relative abundance in both ecotypes and locations; 2) significant presence in the indicator taxon analyses.

Conclusion

27. Ln 457. What do you mean by 'the disconnect' between the plant host ecotype and rhizobiome?

In our research we observed a very strong location effect on the bacterial community richness, diversity and composition. However, we also observed an association between the ecotypes and local soil microbes at their homesite. This association is more visible at the both sides of the extreme conditions like wet Carbondale and dry Hays location, highlighting the mutual recognition between local ecotype and local microbe. We believe it is due to the “home field advantage” and ability of the local ecotype to select and recruit the specialist microbes with the potential beneficial to the plant function. We believe that ecotypes rely on these specialistic microbes, therefore with the coming environmental change, these relationships might be lost. We made changes in the text in the revised manuscript to reflect this idea, and improve clarity for the readers:

Ln 550 - 558: After growing in a common garden for over 10 years, we could still see the distinct microbial communities recruited by the ecotypes. Therefore, these differences across rhizosphere microbiomes recruited by ecotypes across all sites demonstrated that the plant host-rhizobiome interaction is much more exclusive and fragile than we imagined. We believe that ecotypes rely on these relationships with microbes to overcome abiotic stress [105], and with the change in climate, these relationships might be threatened.

References

1. Johnson LC, Olsen JT, Tetreault H, DeLaCruz A, Bryant J, Morgan TJ, et al. Intraspecific variation of a dominant grass and local adaptation in reciprocal garden communities along a US Great Plains' precipitation gradient: implications for grassland restoration with climate change. *Evol Appl.* 2015;8: 705–723.
2. Bender MH, Baskin JM, Baskin CC. Age of Maturity and Life Span in Herbaceous, Polycarpic Perennials. *Bot Rev.* 2000;66: 311–349.
3. Simmons T, Caddell DF, Deng S, Coleman-Derr D. Exploring the Root Microbiome: Extracting Bacterial Community Data from the Soil, Rhizosphere, and Root Endosphere. *J Vis Exp.* 2018. doi:10.3791/57561
4. Brady NC. *The Nature and Properties of Soils.* Macmillan; 1974.
5. Wei Z, Hu X, Li X, Zhang Y, Jiang L, Li J, et al. The rhizospheric microbial community structure and diversity of deciduous and evergreen forests in Taihu Lake area, China. *PLoS One.* 2017;12: e0174411.
6. Hayat R, Ali S, Amara U, Khalid R, Ahmed I. Soil beneficial bacteria and their role in plant growth promotion: a review. *Ann Microbiol.* 2010;60: 579–598.
7. Wagg C, Bender SF, Widmer F, van der Heijden MGA. Soil biodiversity and soil community composition determine ecosystem multifunctionality. *Proc Natl Acad Sci U S A.* 2014;111: 5266–5270.
8. Xu Q, Vandenkoornhuysen P, Li L, Guo J, Zhu C, Guo S, et al. Microbial generalists and specialists differently contribute to the community diversity in farmland soils. *J Adv Res.* 2022;40: 17–27.
9. Ma W, Tang S, Deng Z, Zhang D, Zhang T, Ma X. Root exudates contribute to belowground ecosystem hotspots: A review. *Front Microbiol.* 2022;13: 937940.

June 9, 2023

Dr. Sonny T.M. Lee
Kansas State University
Division of Biology
Manhattan

Re: Spectrum00208-23R1 (Home-field advantage affects the local adaptive interaction between *Andropogon gerardii* ecotypes and rhizobiome)

Dear Dr. Sonny T.M. Lee:

Link Not Available

Sincerely,

Renee Arias

Journals Department
Reviewer comments:

Since the manuscript focuses on the bacterial population of the root system and not the actual Rhizobiome, this should be reflected both in the title and in the body of the manuscript, please modify accordingly.

Reviewer #2 (Comments for the Author):

I had reviewed a prior version of this manuscript. This version has improved in clarity and detail, in particular the experimental design, including sampling characteristics and number of replicates, and details on software use and some aspects of data analysis. In addition, it incorporates discussion about measured soil characteristics across each site and how these could influence the interaction between site, ecotypes and bacterial taxa. There are a few areas where clarification is still needed, including aspects of the statistical analysis performed. Details are indicated below by section/line number.

Throughout the text I recommend the use of bacterial communities instead of rhizobiome, as the article only analyzed the bacterial component of the rhizobiome. This is important because we expect bacteria and fungi (or other microbes) respond different to environmental stressors and variation. Alternatively, the text of the manuscript should indicate that the conclusions presented might not be the same if the study was looking at other microbial groups, given differences in how they adapt and respond to environment and host.

Ln 126. Add "and" before (2)

Ln 92-93. This statement could be revised to indicate that this type of work has not been done for natural ecosystems or plant populations. As in agricultural research there is much research and several examples of host-genotype-environment interactions in microbial community assembly.

Ln 129. For the purpose of this manuscript, it could be helpful to define what "better performance" in rhizosphere recruitment is. Is this measure by greater number of unique taxa, higher bacterial richness or evenness?

Ln 170-171. Fix the symbols in the parenthesis.

Ln 177. There is inconsistency in the total number of samples analyzed for microbial communities. If 7 out of 360 plants died, then there were more than 284 samples for DNA extraction and library preparation. It seems like the 284 samples were the final analyzed after sequence drop-outs and low quality samples removed.

Ln 201. Where does the number 278 comes from? Were the soils collected form the original collection sites of the planting sites.

Ln 224. What was the model used for statistical analysis? Indicate in the text.

Ln 248-267. This section (and in the response to the reviewers) the authors indicate that the main and interactive effects were analyzed. In the results section, including tables, there is no indication about these results, including if the interactions were (or not) significant. These results are important for this study, therefore they should be included in the text (even if the results were non-significant).

Ln 294 - 347. This section discusses differences in soil characteristics across site. Indeed, site is the main driver of soil chemistry differences- which also is related to a precipitation gradient. However, I don't think the statement that "Differences in precipitation result in differences with soil chemistry" is correct. I suggest to rewrite the subtitle.

Ln 326. I think it could be important to indicate why differences in soil measurements in relation to ecotype, are important. For example, could these reflect effects of leaf litter and different in composition of these litter (and their accumulation) by different ecotypes.

Ln 566. Font size is different. Revise throughout the text.

Fig 4- inconsistent font size. Also, the figure legend indicates genera, but in the figure the names are listed at the family level. Be consistent in the taxonomic units used. Also, for the Venn diagram analysis, it is possible to run a chi-test or some other analysis to determine if the greater number of unique out's in the matching ecotype is different than chance? This could be incorporated in the text.

Figure 5. Do the result of the oligotype analysis show home field advantage?

Supplementary files: Make sure that in the final version the order of the supplementary information files is clear.

Staff Comments:

Preparing Revision Guidelines

- Point-by-point responses to the issues raised by the reviewers in a file named "Response to Reviewers," NOT IN YOUR

COVER LETTER.

- Upload a compare copy of the manuscript (without figures) as a "Marked-Up Manuscript" file.
- Each figure must be uploaded as a separate file, and any multipanel figures must be assembled into one file.
- Manuscript: A .DOC version of the revised manuscript
- Figures: Editable, high-resolution, individual figure files are required at revision, TIFF or EPS files are preferred

Please return the manuscript within 60 days; if you cannot complete the modification within this time period, please contact me. If you do not wish to modify the manuscript and prefer to submit it to another journal, please notify me of your decision immediately so that the manuscript may be formally withdrawn from consideration by Microbiology Spectrum.

Dear Dr. Renee Arias
Editor, Microbiology Spectrum

We are thankful to the reviewer #2 for their suggestions and comments. We have made changes to the manuscript (Revised_Andropogon_16S_SPIPS.doc) based on their comments (Response to Reviewers.pdf). Briefly, we

- Adjusted our language and added details to ensure clarity of our data analyses.
- We also added the additional Pearson's chi-square test for the clarity of the Venn diagram.

We have included a revised "Revised_Andropogon_16S_SPIPS.doc" and "Marked-Up Manuscript" version for your perusal. Our responses to the reviewers' comments are outlined in "blue". The line numbers mentioned here correspond to the source text file of the "Revised_Andropogon_16S_SPIPS.doc" - clean version. We hope that you and the reviewers will agree that changes to our manuscript provide more clarity of our work to the readers of Microbiology Spectrum.

Reviewer #2

1. Throughout the text I recommend the use of bacterial communities instead of rhizobiome, as the article only analyzed the bacterial component of the rhizobium. This is important because we expect bacteria and fungi (or other microbes) to respond differently to environmental stressors and variation. Alternatively, the text of the manuscript should indicate that the conclusions presented might not be the same if the study was looking at other microbial groups, given differences in how they adapt and respond to environment and host.

We agree with the reviewer's comment about the differences between the rhizobium and the root-associated bacterial communities. We have made changes in the text in the revised manuscript.

2. Ln 126. Add "and" before (2)

The manuscript has been updated to the following (Ln 138):

Here, we (1) investigated the relative importance of the environment and ecotypic variation of *A. gerardii* on establishing the plant root-associated bacterial communities;

and (2) compared the abilities of three regional *A. gerardii* ecotypes planted reciprocally, to recruit microbes in local and non-local environments. We hypothesized that *A. gerardii* ecotypes would perform better in rhizosphere microbial recruitment at the site closely matching their “home” environment, highlighting the effect of the plant genetic background or ecotype on rhizosphere community assembly.

3. Ln 92-93. This statement could be revised to indicate that this type of work has not been done for natural ecosystems or plant populations. As in agricultural research there is much research and several examples of host-genotype-environment interactions in microbial community assembly.

The manuscript has been revised with the following sentence (Ln 93 - 95):

Although there are numerous studies on plant host influence on root-associated bacterial communities composition, little information is available in the natural ecosystems about the interactive impact of the host-environment on these communities.

4. Ln 129. For the purpose of this manuscript, it could be helpful to define what "better performance" in rhizosphere recruitment is. Is this measure by greater number of unique taxa, higher bacterial richness or evenness?

We thank the reviewer for the comment. We have corrected the wording and indicated that the better performance in our paper indicates the recruitment of the unique microbial populations at the “homesite” of the plant host. We made correction to the following sentence in the revised manuscript (Ln 140 - 143):

We hypothesized that *A. gerardii* ecotypes would perform better in rhizosphere microbial recruitment of unique microbes at the site closely matching their “home” environment, highlighting the effect of the plant genetic background or ecotype on rhizosphere community assembly.

5. Ln 170-171. Fix the symbols in the parenthesis.

We had fixed the symbols in the following text in the revised manuscript (Ln 193):

However, the plant mortality differed in the Pearson’s chi-squared test neither for ecotype (χ^2 -squared = 4.44, df = 2, p-value = 0.109) nor population (χ^2 -squared = 9.43, df = 12, p-value = 0.665), suggesting that mortality was random without predictable patterns across ecotypes or populations.

6. Ln 177. There is inconsistency in the total number of samples analyzed for microbial communities. If 7 out of 360 plants died, then there were more than 284 samples for

DNA extraction and library preparation. It seems like the 284 samples were finally analyzed after sequence drop-outs and low quality samples were removed.

We are really appreciative of the thorough review of the text by the reviewer. We indeed extracted DNA from all 353 samples. And the samples were removed from the analysis later in the process due to low-quality (as indicated in Ln 213 - 219). We have revised the following statement in the manuscript to reflect the right numbers of samples (Ln 200):

We extracted total DNA from 0.150 g of roots and rhizosphere soil from all 353 samples using an Omega E.Z.N.A. Soil DNA Kit (Omega Bio-Tek, Inc., Norcross, GA, USA) as per the manufacturer's protocol with a slight modification.

7. Ln 201. Where does the number 278 come from? Were the soils collected from the original collection sites of the planting sites.

We understand the reviewer's confusion about the number of samples used for the soil chemistry analysis. Yes, the total of 360 samples were collected from the three reciprocal gardens. However, after aliquoting samples for the DNA extraction and manually picking out the root particles, it was not enough material for the soil chemistry lab to robustly produce accurate soil chemistry results. Therefore, only 278 samples were used for our soil chemistry analysis. Regardless of the lower number of samples being processed for soil chemical analyses, we did not find a pattern among those samples. The following was in the revised manuscript (Ln 222 - 234):

We performed soil %C, %N analysis on an aliquot of rhizosphere soil samples. For soil chemistry, the rhizosphere soil samples were homogenized from each soil core (total n = 360) through a 4-mm sieve to homogenize the soil and remove rocks and large pieces of roots, and followed by handpicking small roots from each soil sample. Due to the abundance of root material in the sampled cores, the weight of some soil samples were not enough to accurately measure the soil chemistry. We did not observe the pattern across low weight samples and therefore removed them from following processing resulting in a total of 278 samples used for the soil chemistry.

8. Ln 224. What was the model used for statistical analysis? Indicate in the text.

The details about the model were added to the text in the revised manuscript (Ln 331 - 332):

Our overall analysis (with site, ecotypes, populations nested within populations and blocks as factors) highlighted that site was the main driver of the soil chemistry parameters (ANOVA, pH: $F_{2,27} = 226.31$; $P < 0.001$; Moisture content: $F_{2,27} = 124.90$; $P < 0.001$) (Figure 1, Table 1, Supplementary Table S4).

9. Ln 248-267. This section (and in the response to the reviewers) the authors indicate that the main and interactive effects were analyzed. In the results section, including tables, there is no indication about these results, including if the interactions were (or not) significant. These results are important for this study, therefore they should be included in the text (even if the results were non-significant).

We agree with the reviewer that the interaction terms are important and even not significant results can provide a lot of insight into understanding the generated data. Yes, the interaction terms were always included in the overall and split by site models. We did observe the interaction terms between site and ecotype across community richness and phylogenetic diversity in the overall model. However, the pairwise comparisons after the adjustment for multiple comparisons, did not reveal any significant combination. We believe that those differences were driven by the overpowering site effect. This observation also contributed to our decision to split the data by site and look directly at the ecotype effect at each site separately. The interaction term analyses have been added to the revised manuscript (Ln 402 - 407):

Apart from the main effects, we also observed some significant interaction terms in the overall model. Although, the significant interaction terms were contributed to the location and ecotype effects in community richness and phylogenetic diversity (ANOVA, S_{Obs} : $F_{10,245} = 2.83$; $P = 0.025$; Faith's PD index: $F_{10,245} = 2.72$; $P = 0.0304$), our pairwise comparison tests did not identify the significant combinations (Pairwise TukeyHSD, S_{Obs} : $P_{\text{adj}} > 0.283$; Faith's PD index: $P_{\text{adj}} > 0.280$).

10. Ln 294 - 347. This section discusses differences in soil characteristics across sites. Indeed, site is the main driver of soil chemistry differences- which also is related to a precipitation gradient. However, I don't think the statement that "Differences in precipitation result in differences with soil chemistry" is correct. I suggest rewriting the subtitles.

We thank the reviewer for bringing up this important point. We do agree that it is not in scope of our research to make a conclusion statement on the effect of precipitation of the soil chemistry. However, the main idea of the section is to show that the differences in the site location in perspective to precipitation gradient were associated with distribution of the soil chemistry parameters. We used the word "associated" instead of "result" to better reflect our data in the revised manuscript (Ln 329):

Differences in precipitation are associated with distinct soil chemistry and texture parameters across sites.

11. Ln 326. I think it could be important to indicate why differences in soil measurements in relation to ecotype are important. For example, could these reflect effects of leaf litter and different in composition of these litter (and their accumulation) by different ecotypes. We have added the following statements in the manuscript to reflect the importance of the soil chemistry on the ecotypic (Ln 366 - 370):

The “homesite” soil characteristics play an important role in plant adaptation to the environment and ecotypic divergence [37]. Therefore, the differences in the ecotypic physiology such as root length and litter deposit biomass that varied across ecotypes even when growing at the same site, could potentially result in differences of the microbial recruitment [38,41]

12. Ln 566. Font size is different. Revise throughout the text.

The font in the original document was 10.5 instead of 11. It is fixed in the revised manuscript (Ln 630):

The intraspecific genome content may be particularly important in understanding these host-microbe interactions.

13. Fig 4- inconsistent font size. Also, the figure legend indicates genera, but in the figure the names are listed at the family level. Be consistent in the taxonomic units used.

We thank the reviewer for the comment and paying attention to the details. We have fixed the inconsistencies in the font size and in Fig 4 and corrected the taxonomic units in the figure legend (Ln 996 - 1001)

Figure 4. Venn diagrams representing the overlapping and unique ASVs (numbers in the circle) among (A) all locations, (B) all ecotypes, (C) ecotypes in Carbondale, ecotypes in Hays, ecotypes in Manhattan. The circle with a red edge represents the home ecotype for that location. Inserts represent the top three families. Partial information on unique ASVs are shown here, full information is in Supplementary Table S6. Higher numbers of unique ASVs suggest the “*home field advantage*” of the *Andropogon gerardii* ecotypes.

14. Also, for the Venn diagram analysis, it is possible to run a chi-test or some other analysis to determine if the greater number of unique out's in the matching ecotype is different than chance? This could be incorporated in the text.

We run the chi-square test to check for group independence. As a results the microbes representing sites and ecotypes in Venn diagram were uniques (χ^2 -squared = 953.22, df =

4317, p-value = 1). We have added the the following statistical analysis results in the manuscript (Ln 519 - 521):

To confirm for the independence of the unique microbe group assignments we additionally performed the Pearson's chi-square test for both site ($P > 0.05$) and ecotypes ($P > 0.05$).

Figure 5. Do the results of the oligotype analysis show home field advantage?

The reviewer is correct, the oligotyping analysis may not directly support the “home field advantage” idea, but it definitely highlights two important details that we wanted to point out in our manuscript. First, ecotypes differ in the recruitment not only of the bacterial community, but also on the finer level of microbial oligotypes/strains, that potentially could differ in their functionality. Second, the community level analysis alone is not robust enough to identify differences. We made revisions in the text to reflect these points without focusing too heavily on the “home field advantage” effect. We made changes in the following statements in the manuscript (Ln 625 - 631):

The differences in the Pseudomonadales-Oligos and Rhizobiales-Oligos across the sites potentially highlight the tendency of plants to recruit microbial strains that could contribute to the plant hosts' ecotypic resilience [21,100]. The intraspecific genome content may be particularly important in understanding these host-microbe interactions.

15. Supplementary files: Make sure that in the final version the order of the supplementary information files is clear.

Supplementary files are checked to ensure proper order.

July 5, 2023

Dr. Sonny T.M. Lee
Kansas State University
Division of Biology
Manhattan

Re: Spectrum00208-23R2 (Home-field advantage affects the local adaptive interaction between *Andropogon gerardii* ecotypes and root-associated bacterial communities)

Dear Dr. Sonny T.M. Lee:

Your manuscript has been accepted, and I am forwarding it to the ASM Journals Department for publication. You will be notified when your proofs are ready to be viewed.

Sincerely,

Renee Arias
Editor, Microbiology Spectrum